# IT-SNOW: a snow reanalysis for Italy blending modeling, in-situ data, and satellite observations (2010-2021)

Francesco Avanzi[1], Simone Gabellani[1], Fabio Delogu[1], Francesco Silvestro[1], Flavio Pignone[1], Giulia Bruno[1,2], Luca Pulvirenti[1], Giuseppe Squicciarino[1], Elisabetta Fiori[1], Lauro Rossi[1], Silvia Puca[3], Alexander Toniazzo[3], Pietro Giordano[3], Marco Falzacappa[3], Sara Ratto[4], Hervé Stevenin[4], Antonio Cardillo[5], Matteo Fioletti[6], Orietta Cazzuli[6], Edoardo Cremonese[7], Umberto Morra di Cella[1,7], and Luca Ferraris[1,2]

[1]CIMA Research Foundation, Via Armando Magliotto 2, 17100 Savona, Italy
[2]DIBRIS University of Genoa, Genova, 16145, Italy
[3]Italian Civil Protection Department, Rome, Italy
[4]Regione Autonoma Valle d'Aosta, Centro funzionale regionale, Via Promis 2/a, 11100 Aosta, Italy
[5]Molise Region, Civil Protection, Regional Functional Center, Campochiaro (Cb) - Italy
[6]Environmental Protection Agency of Lombardia, Milano, Italy
[7]Environmental Protection Agency of Aosta Valley, Loc. La Maladière, 48-11020 Saint-Christophe, Italy

**Correspondence:** Francesco Avanzi (francesco.avanzi@cimafoundation.org)

**Abstract.**

We present IT-SNOW, a serially complete and multi-year snow reanalysis for Italy ($\sim$301k km$^2$) – a transitional continental-to-Mediterranean region where snow plays an important, but still poorly constrained societal and ecological role. IT-SNOW provides $\sim$500-m, daily maps of Snow Water Equivalent (SWE), snow depth, bulk-snow density, and liquid water content for the initial period 01/09/2010 - 31/08/2021, with future updates envisaged on a regular basis. As the output of an operational chain employed in real-world civil-protection applications (S3M Italy), IT-SNOW ingests input data from thousands of automatic weather stations, snow-covered-area maps from Sentinel-2, MODIS, and H-SAF products, and maps of snow depth from the spazialization of over 350 on-the-ground snow-depth sensors. Validation using Sentinel-1-based maps of snow depth and a variety of independent, in-situ snow data from three focus regions (Aosta Valley, Lombardia, and Molise) shows little to no mean bias compared to the former, and Root Mean Square Errors on the typical order of 30 to 60 cm and 90 to 300 mm for in-situ, measured snow depth and Snow Water Equivalent, respectively. Estimates of peak SWE by IT-SNOW are also well correlated with annual streamflow at the closure section of 102 basins across Italy (0.87), with ratios between peak water volume in snow and annual streamflow that are in line with expectations for this mixed rain-snow region (22% on average and 12% on median). Examples of use allowed us to estimate $13.70 \pm 4.9$ Gm$^3$ of water volume stored in snow across the Italian landscape at peak accumulation, which on average occurs on the $4^{th}$ of March $\pm 10$ days. Nearly 52% of mean seasonal SWE is accumulated across the Po river basin, followed by the Adige river (23%), and central Apennines (5%). IT-SNOW is freely available at the following DOI: https://doi.org/10.5281/zenodo.7034956 (Avanzi et al., 2022b) and can contribute to better constraining the role of snow for seasonal to annual water resources – a crucial endevor in a warming and drier climate.

# 1 Introduction

The seasonal snow cover is a key modulator of global climate (Flanner et al., 2011) and a primary source of freshwater for more than one sixth of the world population (Barnett et al., 2005; Immerzeel et al., 2020). Snow water resources play a particularly important role in Mediterranean, summer-dry regions, where winter accumulation and the following summer freshet provide highly needed runoff to support societies and ecosystems as their demand peaks and precipitation declines (Zanotti et al., 2004; Bales et al., 2006; Viviroli et al., 2007). Snow-dominated regions include, among others, life hotspots like the US Mountain West, where snow provides 53% of total runoff (Li et al., 2017), Central Asia (Immerzeel et al., 2010), the Andes (Soruco et al., 2015), and the European Alps (Viviroli et al., 2007), from where benefits of snow propagate downstream and across the globe (Sturm et al., 2017). The critical role of snow for water resources, energy management, and ecosystem services is at the foundation of one of the most recurring, simplest, and yet most elusive questions in mountain hydrology (Bales et al., 2006; Margulis et al., 2015; Sturm et al., 2017): how much snow is accumulated across the landscape at any given time?

Despite major advances since the seminal 1906 field campaigns by Dr. James E. Church on Mount Rose (NV, USA), this quest for quantifying snow amount and distribution remains wide open (Dozier et al., 2016). In-situ measurements from ultrasonic snow depth sensors (Ryan et al., 2008) or snow pillows (Cox et al., 1978) are only representative of point conditions, with extrapolation at larger scales being hindered by the striking spatial heterogeneity of the snowpack (Grünewald et al., 2010; Grünewald and Lehning, 2015; De Michele et al., 2016) and possible perturbations of in-situ instrumentation to snow natural conditions (Malek et al., 2017). Extrapolation may be assisted by measuring snow amount along courses (Rice and Bales, 2010), or at strategically chosen locations that are representative of large-scale patterns (Zhang et al., 2017b), but these solutions still imply intense labor and a comparatively high budget. Remote sensing, whether in the form of airborne lidar (Kirchner et al., 2014; Painter et al., 2016), remotely-piloted aircrafts (Bühler et al., 2016; De Michele et al., 2016; Harder et al., 2016; Avanzi et al., 2018), or optical and microwave satellites (Dietz et al., 2012; Gascoin et al., 2019b), has recently gained positions in this context, particularly because it allows one to capture the full spatial distribution of the snowpack (Blöschl, 1999; Lievens et al., 2019). However, remote sensing techniques are limited by either comparatively long revisit times, small areal coverage, uncertainties related to complex morphology, high maintenance costs, or cloud coverage. Finally, snowpack distributed models can simulate snow amount at virtually any resolution, but uncertainties in input data and in process representations make estimates solely based on modeling of limited value in operational snow hydrology (Tang and Lettenmaier, 2010; Pagano et al., 2014; Avanzi et al., 2020).

Reanalyses obtained by assimilating in-situ and remote-sensing data into models are progressively becoming the most frequent, and arguably the most successful, solution to estimate snow water resources. Recent examples of such reanalyses for snow are the Snow Water Equivalent (SWE) product by Margulis et al. (2016) across the California Sierra Nevada, the hyper-resolution ensemble-based reanalysis applied in Switzerland by Fiddes et al. (2019), the meteorological and snow reanalysis across the French mountains by Vernay et al. (2021), the High Mountain Asia UCLA Daily Snow Reanalysis by Liu et al. (2021), or the Austrian reanalysis product by Olefs et al. (2020). Estimates of snow coverage and amount are also available through Earth-system reanalyses like the ERA suite by ECMWF (https://doi.org/10.24381/cds.e2161bac, last

access 19/07/2022), the NASA Global Land Data Assimilation System (GLDAS, see https://ldas.gsfc.nasa.gov/gldas, last access 19/07/2022), or the Japanese 55-year Reanalysis (https://jra.kishou.go.jp/JRA-55/index_en.html, last access 19/07/2022), among many others. Despite inheriting some of the original uncertainty in data and models, reanalysis products optimally combine data and models in reconciled estimates and provide a consistent coverage in space and time, thus paving the way for a new generation of snow science.

We present IT-SNOW, a $\sim$500-m snow reanalysis providing estimates of snow patterns across Italy ($\sim$301k km$^2$) – a topographically and climatically complex region including some of the highest peaks in Europe (the Alps and the Apennines) and partially snow-dominated, socio-economically relevant regions like the Po river basin or central Apennines. To our knowledge, this is the first open, sub-kilometric, serially complete, and multi-year snow reanalysis providing information on snow depth and mass for the Italian territory. Thus, IT-SNOW fills an important scale gap between in-situ measurements and climate models or satellite-based datasets at kilometric resolution – such as the already mentioned ERA suite at 9 km, the H-SAF suite (https://hsaf.meteoam.it/Products/ProductsList?type=snow, last access on 19/08/2022), the Twentieth Century Reanalysis Project at $\sim$200 km or more (Compo et al., 2011), the NCAR Climate Forecast System Reanalysis at $\sim$50 km or more (Saha et al., 2014), the NASA MERRA reanalysis product at $\sim$50 km or more (Gelaro et al., 2017), or the non-mountainous Glob-Snow product at 25 km (Pulliainen et al., 2020, see a complete review at https://globalcryospherewatch.org/reference/snow_inventory.php, last access 19/08/2022).

IT-SNOW blends modeling, in-situ data from snow depth sensors, and satellite observations from Sentinel-2, MODIS, and the H-SAF initiatives, and is the output of a real-time operational monitoring chain developed and maintained by CIMA Research Foundation for the Italian Civil Protection Department, S3M Italy (S3M stands for Snow Multidata Mapping and Modeling, the underlying model used in this operational chain). The present dataset (IT-SNOW v1.0) includes daily reanalyzed outputs of SWE, snow depth, density, and bulk liquid water content from S3M Italy for water years 2011 through 2021 (a water year is defined as a period between the $1^{st}$ of September and the following $31^{st}$ of August and is indicated with the calendar year in which it ends). Future updates are planned to expand this dataset (see Section 4). IT-SNOW is freely available at the following DOI: https://doi.org/10.5281/zenodo.7034956 (Avanzi et al., 2022b).

The paper is organized as follows. Section 2 describes S3M Italy (Section 2.1) and the preparation of the IT-SNOW reanalysis over the historical period 01/09/2010 - 31/08/2021 (Section 2.2). Section 3 evaluates the performance of IT-SNOW by using remote-sensing data, in-situ data, and an indirect water-balance approach using streamflow; this Section also includes a discussion of IT-SNOW sources of uncertainty (Section 3.3). Finally, Section 4 provides examples of use, while Section 5 details data format and standards.

## 2 S3M Italy and IT-SNOW

### 2.1 The S3M Italy operational chain

S3M Italy provides real-time, spatially explicit estimates of snow cover patterns at $\sim$200-m resolution and with a latency of a few hours for the whole of the Italian territory ($\sim$301k km$^2$). This operational chain includes algorithms to ingest in-

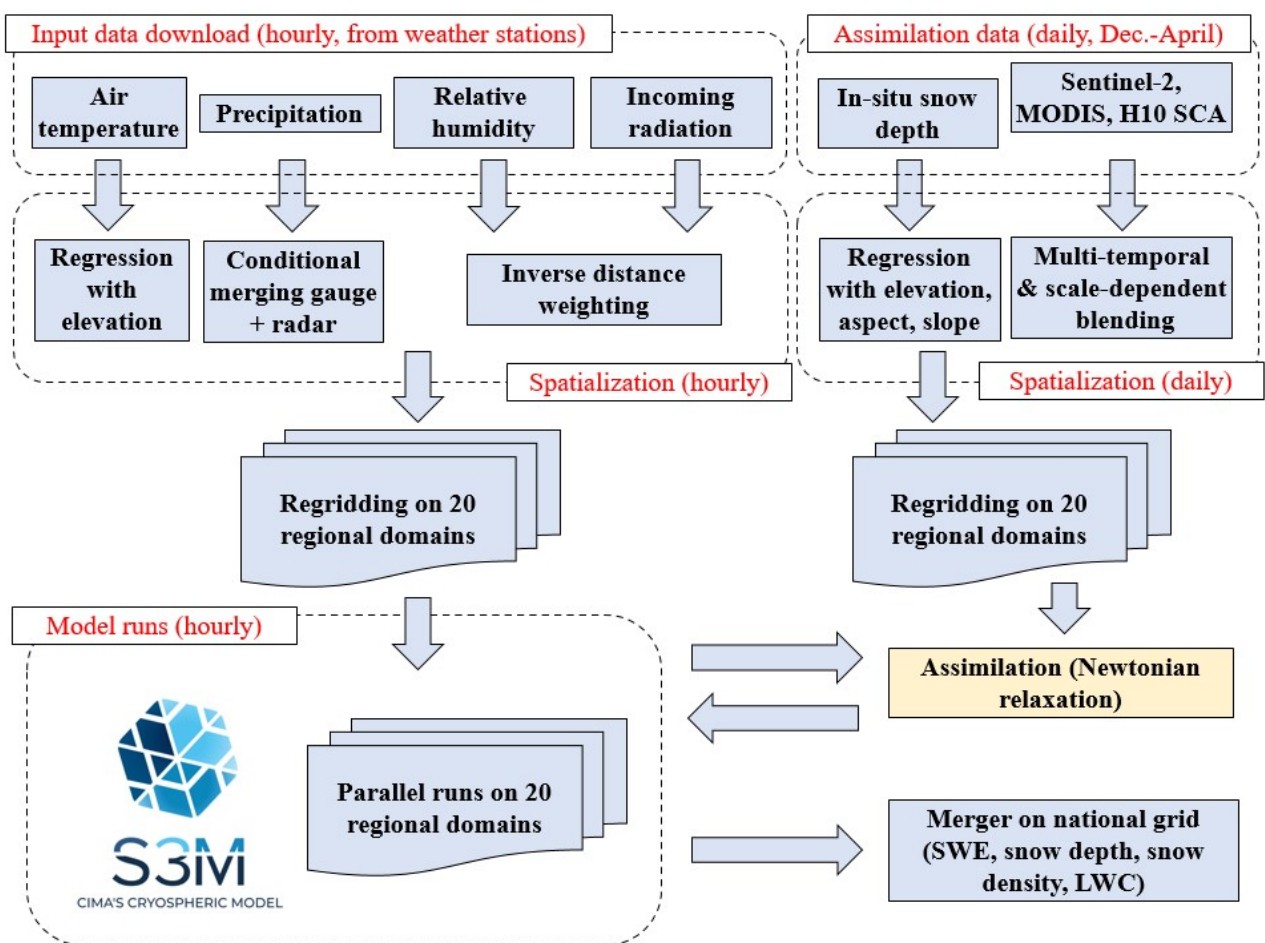

**Figure 1.** Schematic of methods and data flows in S3M Italy, the operational chain used to generate the IT-SNOW dataset. All components of this chain are available in an open source framework at https://github.com/c-hydro/ (last access 30/08/2022). SWE is Snow Water Equivalent, LWC is Liquid Water Content of snow, and SCA is snow covered area.

situ weather station data and satellite maps of snow cover, spatialization and remapping tools to generate weather-input and assimilation maps, parallel scripts to manage model simulations on multi-core servers, as well as a variety of post-processing and maintenance tools to generate final visualizations. S3M Italy is open source and available at https://github.com/c-hydro/ (last access 30/08/2022), in particular through the Python package called *fp-s3m*. A schematic of methods and data flows of this operational chain is reported in Figure 1.

The core of S3M Italy (and hence of the reanalysis IT-SNOW) is the Snow Multidata Mapping and Modeling system, or S3M (Avanzi et al., 2022a). S3M is a spatially distributed cryospheric model solving the snow mass balance and parametrizing snow melt using a hybrid, temperature-index and radiation-driven melt approach. Other processes included in S3M are snow settling, liquid-water outflow, snow-albedo evolution, and precipitation phase partitioning. Complementary to these processes

is an estimate of glacier melt based on the same hybrid approach used for snow, but with modified parameters. Land-cover effects, turbulent fluxes, and snow-forest interactions are currently not taken into account.

S3M is a raster-based model, where snow model equations are solved for each cell with no exchange of mass or energy across pixels. S3M is also open source and freely available at https://github.com/c-hydro/s3m-dev (last access 30/08/2022), while more details on model physics and user requirements can be found in Avanzi et al. (2022a).

### 2.1.1 Input data preparation


Every hour at HH:40, input data required by the model are downloaded and saved in pre-defined formats. These inputs include total precipitation, air temperature, relative humidity, and solar radiation, all of which are obtained from the database of the Italian Regional Administrations, Autonomous Provinces, and the Italian Civil Protection. Input data have an hourly time step. To fill potential gaps due to occasional malfunctioning and/or failures, every hour automatic procedures check the existence

of hourly inputs for the last 30 hours. An unique estimate of the precision of considered weather data is not available as the type of sensor installed varies from one region to another. The installation and the maintenance of the sensors generally follow guidelines from the World Meteorological Organization, to which the reader is referred (WMO, 2018).

Total precipitation fields are the result of a modified conditional merging approach applied to precipitation gauges (spatial density of $\sim 1/100$ km$^2$) and radar observations (Bruno et al., 2021), so no further spatialization is performed in S3M Italy

(Figure 2). This modified conditional merging spatializes in-situ precipitation data using an approach similar to Kriging (called GRISO, from the Italian version of Random Generator of Spatial Interpolation from Uncertain Observations) where, however, the covariance structure is estimated for each precipitation gauge and each hour using radar data (see full details in Sinclair and Pegram, 2005; Apicella et al., 2021; Bruno et al., 2021; Lagasio et al., 2022). Final maps have a resolution of $\sim 1$ km$^2$, with a median Root Mean Square Error of less than 1 mm for a selection of 70 heavy precipitation events in Italy with accumulation

greater than 100 mm or maximum precipitation rate greater than 50 mm/h during the 2011–2014 period (see details in Bruno et al., 2021). No phase partitioning is performed at this stage: separation between rainfall and snowfall is performed by the S3M model using the parametric approach by Froidurot et al. (2014), which relies on both air temperature and relative humidity.

Data of air temperature, solar radiation, and relative humidity are obtained as in-situ point station and further spatialized between HH:50 and HH+1:10 ($\sim 1$ km for temperature and $\sim 500$ m for radiation and relative humidity). For air temperature,

spatialization is performed by organizing station data into meteorological homogeneous regions as dictated by the Italian Civil Protection (see an example in Figure 2, 2019 update) and fitting region-specific hourly linear regressions between air temperature and elevation (Figure 2 and Figure 3). These linear regressions are then applied using the meteorological homogeneous region's Digital Elevation Model to derive temperature maps. Figure 3 reports monthly quartiles and the frequency distribution of daily average national lapse rates as derived through this procedure, which agree with estimates by Rolland (2003) in the

Alps. As for relative humidity and incoming shortwave radiation, we currently employ a computationally efficient method based on inverse distance weighting; no shadow effect or reflections from surrounding terrain are currently considered here, unless these are already captured by the comparatively dense network of stations.

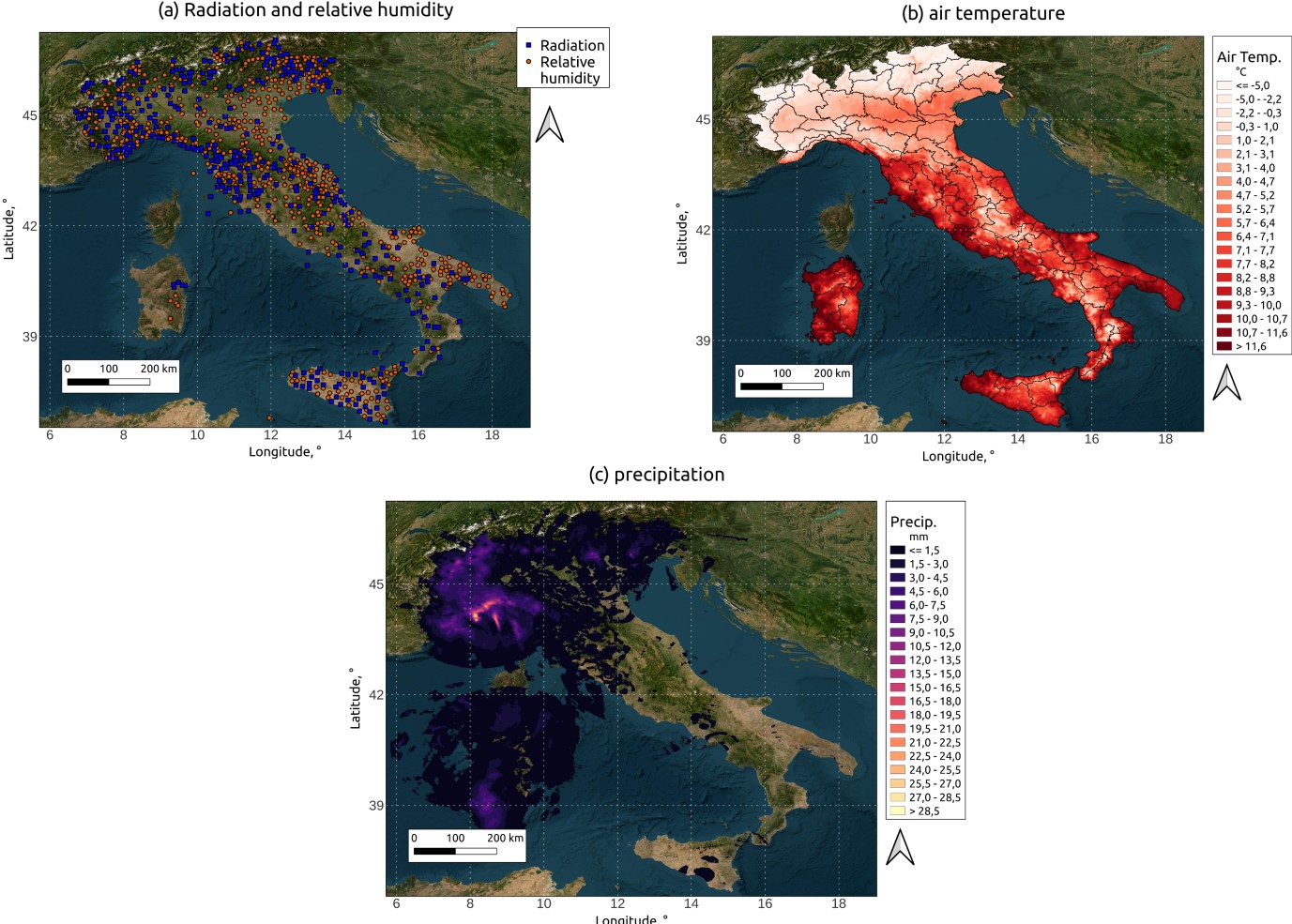

**Figure 2.** Examples of input data used by S3M Italy to produce the IT-SNOW reanalysis. (a): location of radiation and relative humidity sensor stations for 04/02/2021 at 10 a.m. UTC; (b): air temperature map for 14/02/2022 at 08 p.m. UTC, along with delineation of meteorological homogeneous regions in black (2019 update); (c): precipitation map based on a modified conditional merging between precipitation gauges and radars for 23/11/2019 at 09 a.m. UTC (Bruno et al., 2021). Background map: ESRI Satellite theme. Note that some stations in panel (a) may host both types of measurements.

Input data preparation ends between HH+1:10 and ∼ HH+1:20, when input maps are cropped over the 20 computational domains, each corresponding to one Italian administrative region. Computational grids for these 20 domains were originally derived from a 20-m Digital Elevation Model provided by the Italian Institute for Environmental Protection and Research (ISPRA), which was resampled at 200 m resolution using an averaging method. Besided elevation, S3M Italy employs static glacier maps from the Randolph Glacier Inventory v 6.0 (Pfeffer et al., 2014).

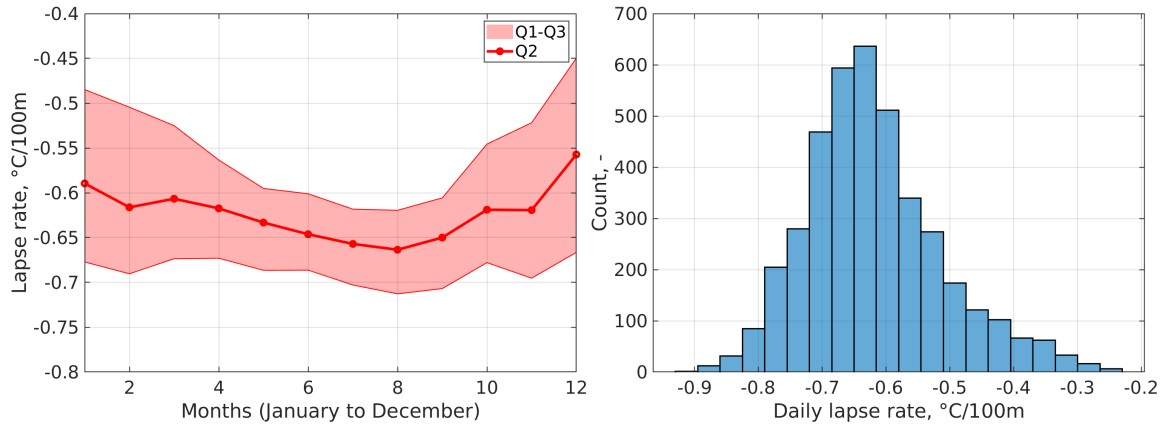

**Figure 3.** Daily national lapse rate climatology (left) and frequency distribution of daily national lapse rates according to S3M Italy – period September 2010 - August 2021. Q1, Q2, and Q3 are the first, second, and third quartiles of daily national lapse rates for each month.

### 2.1.2 Assimilation data preparation

Data assimilation in S3M Italy is performed in the form of both satellite snow covered area (SCA) and snow-depth maps (Figure
4). Maps of snow covered area are produced, once per day, by blending images from the ESA Sentinel-2, the NASA MODIS, and the Eumetsat H-SAF initiatives (product H10). First, 20-m maps of Sentinel-2 for the last 6 days across Italy are mosaicked (using the most recent one in case of overlapping images). A latency of 6 days is allowed both to manage the fairly infrequent revisit time of Sentinel-2 (∼5 days in Italy) and as a first provision to manage cloud obstruction. These maps at 20 m are then resampled at the ∼200-m grid of S3M Italy using the Python raster processing package Rasterio with a mode resampling
approach, so to assign the dominant land cover among the 20 m pixels inscribed in each 200-m pixel. Once this first-guess SCA map is available, cloud-covered or unclassified pixels are further filled using resampled MODIS (https://modis.gsfc.nasa.gov/data/dataprod/mod10.php, last access 13/12/2022) and H-SAF H10 data (https://hsaf.meteoam.it/Products/Detail?prod=h10, last access 13/12/2022), which have both nominal daily frequency and are used as distributed by the respective providers with no further processing. The result is a blended snow map providing information on snow cover, bare ground, and non classified
pixels. Besides mosaicking maps from multiple sources with different revisit times, no additional gap-filling for cloud coverage is performed. Assimilation for cloud-covered pixels was thus foregone. Like input data, this map is then remapped across the 20 regional domains to be assimilated.

Currently, Sentinel-2 SCA is produced by operationally applying the Sen2Cor algorithm by ESA (https://step.esa.int/main/snap-supported-plugins/sen2cor/, last access 30/08/2022). Albeit lower in accuracy than snow-specific and high-resolution products
like Theia (Gascoin et al., 2019a), SCA maps derived with Sen2Cor were validated against snow depth sensors at national scale during the average, representative 2020 snow season and showed typical accuracy scores on the order of 0.7 to 0.8, as expected (see Figure S1 in the Supplement and Main-Knorn et al., 2017).

Snow-depth maps are produced based on the interpolation of snow-depth-sensor in-situ data. Every day during winter, measurements of the ∼350 snow-depth sensors across Italy at 10 a.m. UTC are downloaded and quality-checked using an automatic filtering approach based on seasonality, climatological thresholds on minimum and maximum snow depth, and a filter based on a 6-h moving-window coefficient of variation to detect grass growth after snow melt. The specific device used to automatically monitor snow depth varies across the country, with the majority of them using a ultrasonic principle with an accuracy of ∼ ±1 cm (Ryan et al., 2008).

The ∼350 snow-depth in-situ data are then organized in 10 homogeneous regions defined along the boundaries of the International Standardized Mountain Subdivision of the Alps (SOIUSA, see Valt et al., 2018) based on a tradeoff between maximizing data availability for each region and complying with expected climatology (e.g., we differentiated between inner-Alpine valleys and coastal, maritime mountain ranges, see Figure 4 for a delimitation of these regions). For each of these homogeneous regions, a separate multilinear-regression model is fitted across observed snow depth at sensor locations (predictand), elevation, slope, and aspect (predictors, with slope and aspect retained only if statistically significant). By applying the resulting (daily) multilinear regressions using a Digital Elevation Model of each homogeneous region, daily snow-depth maps are created and then cropped across the 20 computational domains of S3M. This is only done if at least 10 observations are available in a given homogeneous region; otherwise, spatialization and thus assimilation for that homogeneous region is foregone. An evaluation of this multilinear-regression model in Aosta valley showed biases on the order of 10 cm compared to avalanche probes, while a comparison with Sentinel-1 snow-depth data at national scale showed typical biases of up to 5 cm and typical Root Mean Square Errors below 10 cm (see Section 3 for details on these evaluation data).

Along with maps of snow depth, the procedure also generates a Kernel map quantifying spatial uncertainty in the multilinear-regression model based on the distance across snow-depth sensors (Avanzi et al., 2021, 2022a). This Kernel is employed to assimilate snow-depth map via a spatially distributed Newtonian-Relaxation approach (also known as Nudging), under the assumption that uncertainty in snow-depth maps will be lower in areas with a denser network of snow depth sensors. When a snow-depth map is available, the assimilation procedure computes pixelwise differences between a-priori SWE and snow-depth-map-based SWE (after conversion of snow depth maps into SWE maps using modeled density); this difference is then added to a-priori SWE via Kernel weighting.

SCA maps are not assimilated directly, but are used to clip snow-free pixels in snow-depth maps before assimilation in the S3M model (thus leaving without snow instances where snow-depth maps estimate no snow but SCA maps observed snow). Both positive and negative differences are assimilated, meaning that assimilation may result in either a decrease or an increase in simulated SWE. To further cope with grass interference in snow-depth-sensor data, the assimilation of snow depth and SCA maps is only performed between December and April, once per day, conventionally at 10 a.m. UTC.

### 2.1.3 Model runs and postprocessing

Upon completion of the input-data remapping component of the modeling chain, parallel runs of S3M are performed (a first batch is launched at HH+1:24 and a second one at HH+1:34). Run time depends on the size of each modeling domain, with all simulations being completely roughly by HH+2:00, hence a latency of less than 2 hours for all domains. A Python wrapper

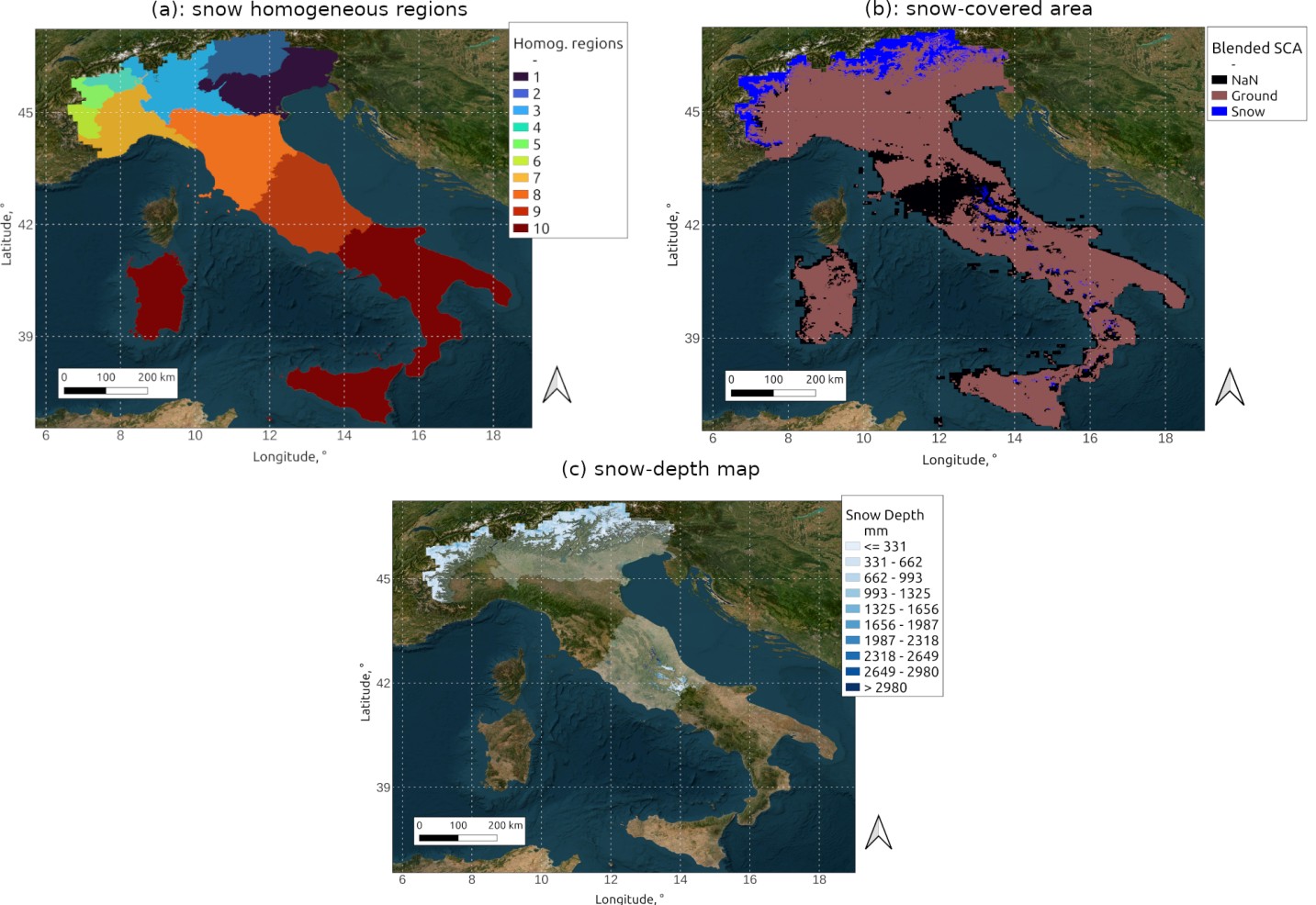

**Figure 4.** Data assimilated in S3M Italy to produce the IT-SNOW reanalysis: (a) homogeneous snow regions based on snow climatology and expert knowledge; (b) blended snow covered area maps based on Sentinel-2 + MODIS + H-SAF H10 snow products for 13/02/2022; (c) snow depth interpolated map for 13/02/2022. For visualization reasons, the snow-depth map in panel (c) was clipped using the concurrent snow covered area map. Snow-depth estimates for some homogeneous regions were missing on that day due to insufficient in-situ data; available regions are showed with a 50% transparency. Background map: ESRI Satellite theme.

manages each run, with S3M being a compiled Fortran executable. Every day at 3 a.m., summaries of previous day's simulations are compiled by mosaicking each domain on a national grid and saving outputs for visualization on the Italian Civil Protection WebGIS maintained by CIMA Foundation, myDewetra.

## 2.2 IT-SNOW preparation

For the scopes of IT-SNOW, we replicated an operational run of S3M Italy over the historical period 01/09/2010 to 31/08/2021 (with a first period from 01/09/2009 through 31/08/2010 used as spin-up). Historically observed weather data were thus downloaded and spatialized as outlined above, while also downloading, processing, and spatializing both satellite SCA maps and snow-depth maps. Note that Sentinel-2 data were used only from summer 2021 and thus assimilated SCA before that period are the result of MODIS + H-SAF maps. The native resolution of this historical run was ∼200 m, in line with the operational chain of S3M Italy.

Outputs of this historical run were saved every 6 hours (5 a.m., 11 a.m., 5 p.m., 11 p.m., all UTC times) and those at 11 a.m. were assumed as representative snapshots of daily conditions. These outputs at 11 a.m. UTC were thus remapped from the native ∼200-m grid to a national, geographic grid at ∼500 m (WGS84, EPSG 4326, pixel size: 0.005057°). No projection was performed to avoid accuracy and distortion issues related to such cartographic systems. Owing to the geographic grid, the actual pixel size in meters changes with latitude, from a minimum of ∼460 m to a maximum of ∼508 m. We remapped at ∼500 m using a nearest neighbor approach for computational-efficiency reasons and as an intermediate trade-off maximizing predictive confidence between the native resolution of S3M Italy and the coarser resolution of precipitation data at 1km$^2$. Remapped outputs include instantaneous Snow Water Equivalent (SWE), snow depth, bulk snow density, and bulk liquid water content, which overall form the IT-SNOW reanalysis dataset (see Section 5). Additional outputs are available upon request, and remapping over alternative grids is also possible.

## 3 IT-SNOW Evaluation

### 3.1 Methods

Given that snow-depth sensor data from the 20 regional networks were assimilated in IT-SNOW, we looked for alternative data that could act as truly independent evaluation sources. The first one was the satellite product by Lievens et al. (2019), C-SNOW, which provides daily, 1-km snapshots of snow depth across mountainous areas of the northern Hemisphere based on an empirical change detection method applied to the Sentinel-1 measurements of the cross-polarization ratio. While C-SNOW is a remote-sensing product and as such may have locally larger uncertainties that in-situ data, it has been successfully compared with in-situ data from ∼4,000 sites, with biases within ±0.1 m for most of them (Lievens et al., 2019). For the scope of evaluating IT-SNOW, the main advantage of C-SNOW compared to in-situ data is that it is natively spatially distributed, and thus allowed us to compare IT-SNOW across the landscape rather than at specific points. The evaluation period went from 01/09/2016 to 08/04/2020, the full span of 1-km C-SNOW data that is currently available. We remapped C-SNOW data onto the ∼500-m grid of IT-SNOW using a nearest-neighbor approach and then computed pixelwise bias and Root Mean Square Error with regard to IT-SNOW. Note that C-SNOW data are available only for dry snow conditions, and that its optimization procedure included some (but not all) of the Italian in-situ snow-depth-sensor data.

The second source of validation data considered here were in-situ data taken in Aosta Valley (north-western Italy, see Figure 6), Lombardia (northern Italy, see Figure 7), and Molise (central Italy, see Figure 8). These three areas present significantly different climates and thus snow types (Sturm and Liston, 2021), with Aosta Valley and Lombardia being characterized by a seasonally consistent and deep Alpine snowpack, and Molise being more exposed to lake-effect snowfalls from the Adriatic Sea and thus to a more ephemeral and maritime snow cover (Da Ronco et al., 2020). These three datasets are topographically diverse and cover a comparatively long time span (see below), meaning that they are representative of larger scale performances of IT-SNOW.

Measurements in Aosta Valley included yearly snow-course manual samples taken at peak accumulation every 50-100 m along elevation transects of several kilometers upstream of five hydropower reservoirs (water years 2011-2021, elevation above 2000 m a.s.l.), daily to weekly manual measurements at recurring locations for avalanche forecasting (water years 2011-2021, elevations above 1000 m a.s.l.), and hourly automatic measurements from a ultrasonic snow depth sensor and a SWE sensor in Torgnon (2012-2020, elevation 2160 m a.s.l., see a data inventory in Avanzi et al., 2021, 2022a). Data in Lombardia include weekly estimates of snow depth and SWE obtained by running the physics-based multi-layer SNOWPACK model (Bartelt and Lehning, 2002) in correspondence of automatic weather and snow stations at medium elevation (1800-2600 m a.s.l.), where the model was forced using local input data and assimilating local snow depth (henceforth, AWS snow depth and SWE – years 2016 through 2021, AWS standing for Automatic Weather Station), and measurements of snow depth and SWE collected between May and June on glacier terrain at very high elevation for mass-balance purposes (elevations above 3000 m a.s.l., years 2016 through 2021, henceforth glacier snow depth and SWE). Data in Molise included daily to weekly manual measurements at four recurring locations for avalanche and water-supply forecasting (2011-2021, elevation 1200-1500 m a.s.l.). Performance metrics between observed and simulated SWE, snow depth, and bulk-snow density included bias, Root Mean Square Error, Mean Absolute Error (MAE), Positive and Negative Mean Error (PME and NME, respectively), the Kling-Gupta Efficiency (Kling et al., 2012), and the Pearson's correlation coefficient.

The third source of validation data were streamflow measurements for a selection of 102 basins in Italy for which long-term, serially complete, and quality-checked time-series of streamflow were available for the period 01/09/2010 through 31/08/2019 (Bruno et al., 2022). We used these data to compare the annual peak of water stored in snow and annual cumulative streamflow at the closure section of these basins (both in $Gm^3$), as a proxy of the proportion of annual flow that was accumulated as snow. To this end, water stored in snow (simply SWE in $Gm^3$ in the following) was obtained by multiplying pixelwise SWE in m w.e. from IT-SNOW by the area of each cell, and then summing all pixelwise values. Given general knowledge of Italian precipitation climatology being a mix between snow and rain, we not only expect these ratios to be between 0 and 1, but also to predominately be smaller than 0.5. Owing to precipitation increasing with elevation (Avanzi et al., 2021), we also expect these ratios to increase with average elevation of each of these catchments. While indirect in that IT-SNOW is not evaluated against snow data, this third evaluation stems from a long-standing tradition of inverting the hydrological cycle (Valery et al., 2009) to provide insights into the consistency of IT-SNOW estimates with the local water budget.

## 3.2 Results

### 3.2.1 Evaluation of snow depth against C-SNOW

Mean pixelwise bias between IT-SNOW and C-SNOW was close to zero (-0.01 m), with a median value of zero and first and third quartiles being -0.03 m and +0.02 m, respectively (Figure 5c). Thus, the distribution of spatial biases was well centered around zero (Figure 5c). Mean pixelwise RMSE was 0.22 m (Figure 5d), that is, close to the mean absolute error found by Lievens et al. (2019) in the original evaluation of C-SNOW with snow depth sensors (0.18 to 0.31 m). First, second, and third quartiles of pixelwise RMSE were 0.14 m, 0.19 m, and 0.27 m, with only a fraction of values above 0.5 m (Figure 5d, both bias and RMSE were calculated between time-series at each pixel). We conclude that the two products provide consistent estimates of snow depth across the Italian landscape.

The spatial distribution of bias across the Italian Alps and the central Apennines showed no obvious pattern, with only a tendency of IT-SNOW to underestimate C-SNOW snow depth at high elevations (see Figure 5a and b). This may be related to well-known biases of precipitation gauges and radars, the main sources of input precipitation in S3M Italy, at high elevations and/or in inner-Alpine areas (Zhang et al., 2017a; Cui et al., 2020; Avanzi et al., 2021, see Section 3.3 for a discussion). Besides this bias with elevation, biases and RMSEs between IT-SNOW and C-SNOW were consistent across the 10 homogeneous regions used to generate snow-depth maps to be assimilated in IT-SNOW (Figure S2 and S3 in the Supplement).

### 3.2.2 Evaluation against in-situ data

**Table 1.** Overview of IT-SNOW performance versus in-situ data in Aosta valley, Lombardia, and Molise. RMSE is the Root Mean Square Error, MAE is the Mean Absolute Error, PME is the Positive Mean Error, NME is the Negative Mean Error, KGE is the Kling-Gupta Efficiency (Kling et al., 2012), and r is the Pearson's correlation coefficient.

| Area | Variable | RMSE | Bias | MAE | PME | NME | KGE | r |
|---|---|---|---|---|---|---|---|---|
| Aosta valley | SWE (Torgnon) | 95 mm | -8.5 mm | 66.2 mm | 59.6 mm | -72 mm | 0.74 | 0.81 |
| | Snow depth (Torgnon) | 30 cm | -1.9 cm | 20 cm | 16.4 cm | -24.7 cm | 0.73 | 0.75 |
| | Density (Torgnon) | 93 kg/m$^3$ | -5.8 kg/m$^3$ | 70 kg/m$^3$ | 59.4 kg/m$^3$ | -82.5 kg/m$^3$ | 0.37 | 0.43 |
| | Snow depth (Aval. probes) | 56 cm | 7.9 cm | 37.6 cm | 37.7 cm | -37.4 cm | 0.25 | 0.55 |
| | Snow depth (courses) | 132 cm | -55 cm | 105 cm | 87 cm | -113 cm | 0.06 | 0.21 |
| Lombardia | SWE (AWS) | 290 mm | -112 mm | 202 mm | 158 mm | -220 mm | 0.54 | 0.57 |
| | Snow depth (AWS) | 63 cm | -19 cm | 43 cm | 39 cm | -44 cm | 0.60 | 0.63 |
| | SWE (glacier) | 842 mm | -717 mm | 722 mm | 139 mm | -733 mm | 0.33 | 0.60 |
| | Snow depth (glacier) | 135 cm | -110 cm | 113 cm | 27 cm | -117 cm | 0.41 | 0.60 |
| Molise | SWE | 200 mm | 82 mm | 135 mm | 161 mm | -80 mm | 0.37 | 0.45 |
| | Snow depth | 61 cm | 34 cm | 44 cm | 49 cm | -26 cm | -0.19 | 0.48 |
| | Density | 109 kg/m$^3$ | -24 kg/m$^3$ | 86 kg/m$^3$ | 82 kg/m$^3$ | -89 kg/m$^3$ | 0.35 | 0.44 |

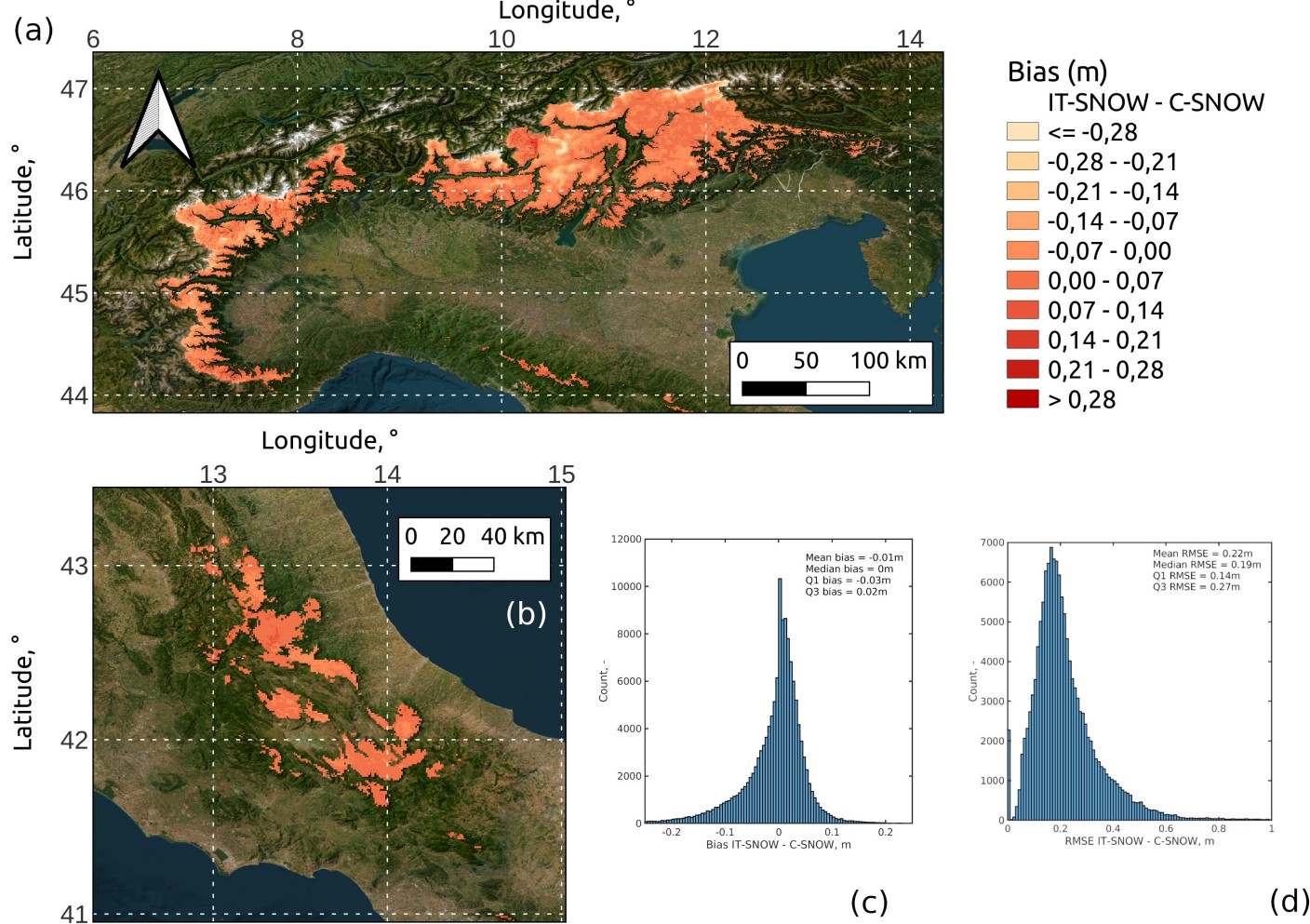

**Figure 5.** Evaluation of IT-SNOW using the Sentinel-1 snow-depth product C-SNOW, period September 2016 through April 2020: pixelwise bias for the Italian Alps (a) and the central Apennines (b), frequency distributions of pixelwise bias (c) and Root Mean Square Error (d). Background map: ESRI Satellite theme. Panel (a) and (b) refer to the two areas of Italy with seasonally deep snow cover.

270    IT-SNOW estimates of snow depth, SWE, and bulk-snow density were generally well correlated with measurements in Torgnon (Aosta valley, Figure 6 e, g, and i), with local RMSE for snow depth, SWE, and density being 30 cm, 95 mm, and 93 kg/m$^3$, respectively (Table 1) and Pearson's correlations between 0.43 and 0.81 (Table 1). Also biases were minor in Torgnon, with Kling-Gupta efficiencies that were significantly above the no-skill threshold of -0.41 (see Knoben et al., 2019, and Table 1). From a seasonal perspective, IT-SNOW and measurements in Torgnon also agreed in terms of accumulation and snowmelt

275    temporal patterns, as well as date of peak accumulation (Figure 6 d, f, and h). The lowest correlation was found for bulk-snow density (Table 1), which is not surprising given that density was indirectly derived from measurements of snow depth and

SWE (DeWalle and Rango, 2011), a procedure that may increase noise (Terzago et al., 2019). Performance scores decreased when considering avalanche probes (Figure 6c and Table 1) and high-elevation snow courses (Figure 6b and Table 1), which was expected given the significant scale mismatch between a ∼500 m reanalysis and topographically diverse, in-situ manual measurements and the already mentioned, possible underestimation of precipitation fields at high elevations (see Figure 5 and Avanzi et al., 2021). Overall, evaluation results in Aosta Valley showed that IT-SNOW successfully reconstructs both the seasonal dynamics and peak timing of snow depth and mass, hence density, of this heavily instrumented region (see Figure 2 and 4).

In Lombardia, IT-SNOW estimates of snow depth and SWE generally agreed well with those provided by SNOWPACK at medium elevations, despite somewhat larger RMSEs and biases, larger Mean Absolute Errors, and a lower correlation than in Aosta valley (Table 1 and Figure 7). Note that these medium-elevation data in Lombardia were not directly observed, but were the result of modeling, which may have increased their own uncertainty. On the other hand, IT-SNOW in Lombardia showed an expected, systematic underestimation of very-high-elevation mass-balance measurements on glaciers (biases of -717 mm and -110 cm, Table 1), despite a promising correlation and KGE between observations and simulations (Table 1). Whilst overcoming these underestimations is a major challenge for large-scale reanalyses like IT-SNOW, again because of the scale and undercatch issues discussed with regard to C-SNOW (Figure 5) and Aosta valley data (Figure 6), and whilst very-high elevation regions play only a minor role in a catchment water balance, we discuss pathways to improve IT-SNOW in this regard in Section 3.3.

Correlations between measured and simulated snow depth, SWE, and density was lower in Molise than in Aosta Valley and Lombardia (Figure 8b, c, and d, RMSE of 61 cm, 200 mm, and 109 kg/m$^3$, respectively, and biases of 34 cm, 82 mm, and -24 kg/m$^3$, respectively – Table 1). We interpret this outcome as due to the sparser network of assimilated snow depth sensors used in southern Italy compared to northern Italy (see Figure 4). Yet, IT-SNOW successfully reconstructed not only the seasonal dynamics of accumulation of melt in Molise, but also the much more significant interannual variability in peak SWE than in Aosta valley (Figure 8f).

These performances of IT-SNOW against in-situ data is consistent with snow reanalyses over other areas of the Alps. In this regard, Vernay et al. (2021) report median RMSEs for snow depth on the order of 10 to 40 cm in France (maximum RMSEs up to 90 cm), with an increasing trend with elevation; both results echo findings in this paper for areas at medium elevation where the bulk of forcing and evaluation data is available (Avanzi et al., 2021). In Austria, Olefs et al. (2020) found RMSEs for snow depth and SWE on the order of 10 cm and 100 mm, with correlations of 0.86 and 0.91, respectively; this accuracy is higher than that of IT-SNOW, likely because of the much more homogeneous coverage of snow data in Austria compared to Italy (see Figure 1 in Olefs et al., 2020). Finally, Fiddes et al. (2019) report RMSEs on the order of 38 to 53 cm for snow depth and 184 to 258 mm for SWE, which again tallies with the accuracy of IT-SNOW.

Simulated time-series of snow depth and SWE in Torgnon present frequent, abrupt oscillations that are not related to any physical process, such as melt or settling (Figure 6d and f). These oscillations, which were already noted in Avanzi et al. (2022a), are due to the assimilated snow-depth maps, which often include – or even propagate – instrumental noise (Ryan et al., 2008). Indeed, similar oscillations are not visible in bulk-snow density time-series, which are unaffected by assimilation (Figure 6h), or in Molise (Figure 8e and f), where assimilation is much rarer due to a sparse network of snow depth sensors.

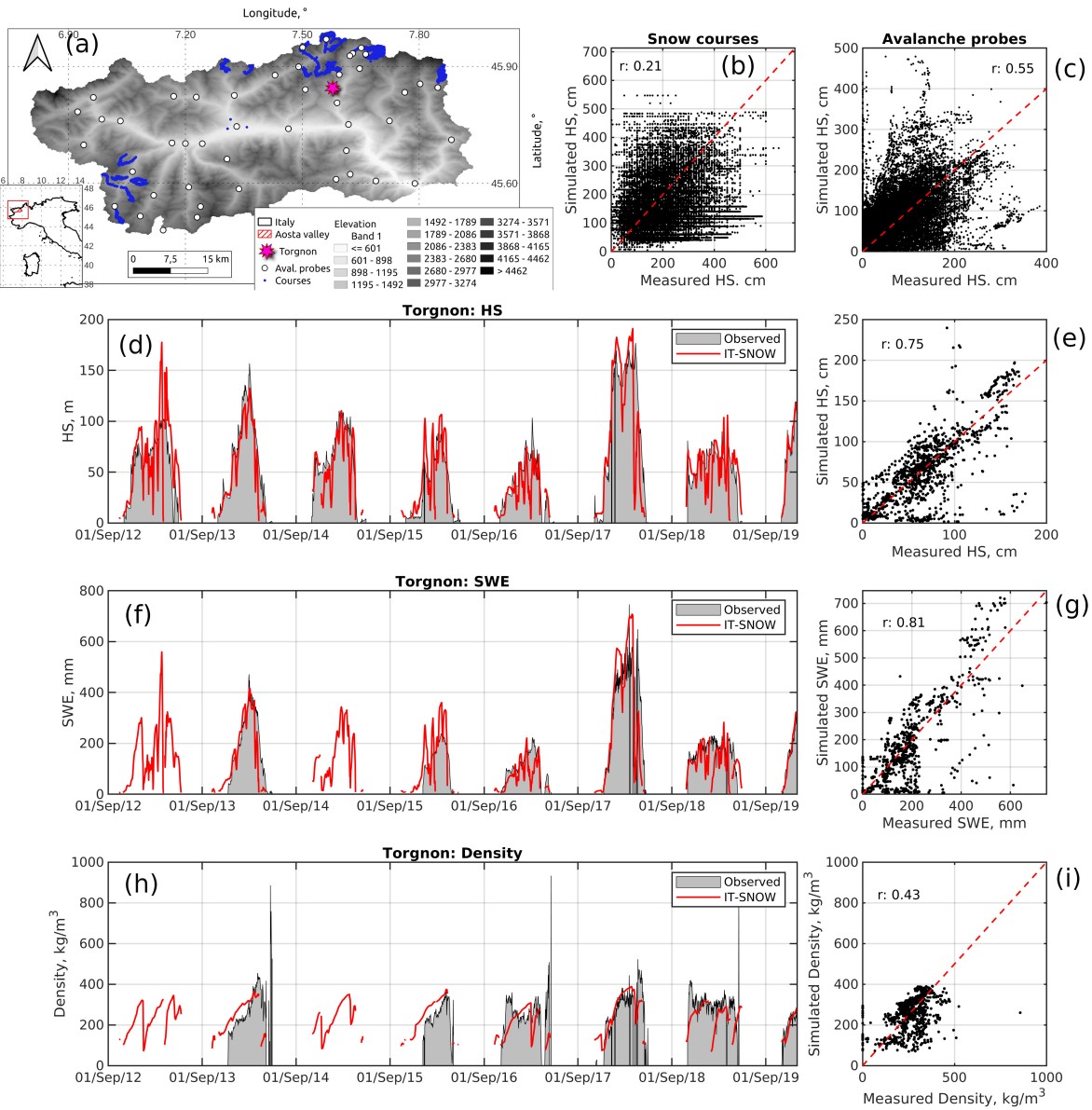

**Figure 6.** Evaluation of IT-SNOW in Aosta Valley. (a): topography of the focus region, along with sampling location of evaluation data; (b) and (c): observed vs. simulated snow depth at snow-course and avalanche-probe locations; (d) and (e): simulated vs. observed snow depth at Torgnon; (f) and (g): simulated vs. observed SWE at Torgnon; (h) and (i) simulated vs. observed bulk-snow density at Torgnon. Note that simulated snow depth, SWE, and density in panels (d), (f), and (h) was smoothed using a 5-day moving window for readibility. r is the Pearson's correlation coefficient, HS is snow depth, and SWE is Snow Water Equivalent.

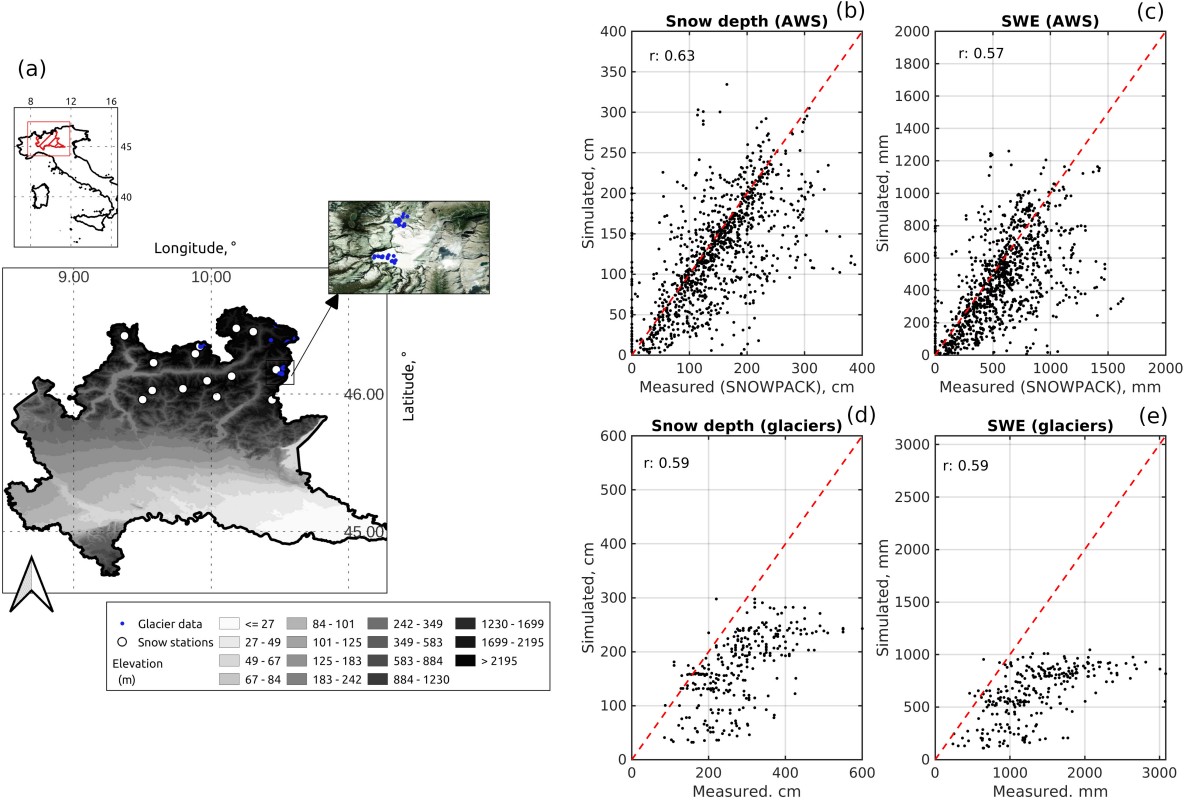

**Figure 7.** Evaluation of IT-SNOW in Lombardia. (a): topography of the focus region, along with location of evaluation data and an example of glacier-data sampling geometry; (b) and (c): IT-SNOW vs. SNOWPACK snow depth and SWE at medium-elevation snow stations, respectively; (d) and (e): observed vs. IT-SNOW snow depth and SWE at very-high-elevation glacier sites in the context of end-of-season mass-balance surveys. r is the Pearson's correlation coefficient, AWS is Automatic Weather Station, and SWE is Snow Water Equivalent.

While these oscillations do not affect the seasonal reconstruction of snow depth, or the temporal patterns of peak SWE, it is important to bear them in mind when performing temporally fine analyses with IT-SNOW. Such oscillations could be better handled by assimilation techniques that explicitly account for point-measurement uncertainty, such as a Kalman or Particle Filter (Piazzi et al., 2018), but doing so would entail significantly higher computational demands that are currently non-feasible given the tight, real-time schedule required by S3M Italy.

### 3.2.3 Comparison of estimated SWE with observed streamflow

On average, peak SWE is about 22% ±24% of annual streamflow across the considered 102 guage stations and 9 water years (median of 12.8%, period 2011-2019, Figure 9b – note that SWE is expressed in this section in the form of total water volume stored in snow, Gm$^3$, see Section 3.1). As expected, the distribution of this ratio is highly skewed toward values below 50%, and shows a clear latitudinal trend, with higher values for rivers draining from the southern Alps in the Po river valley and a

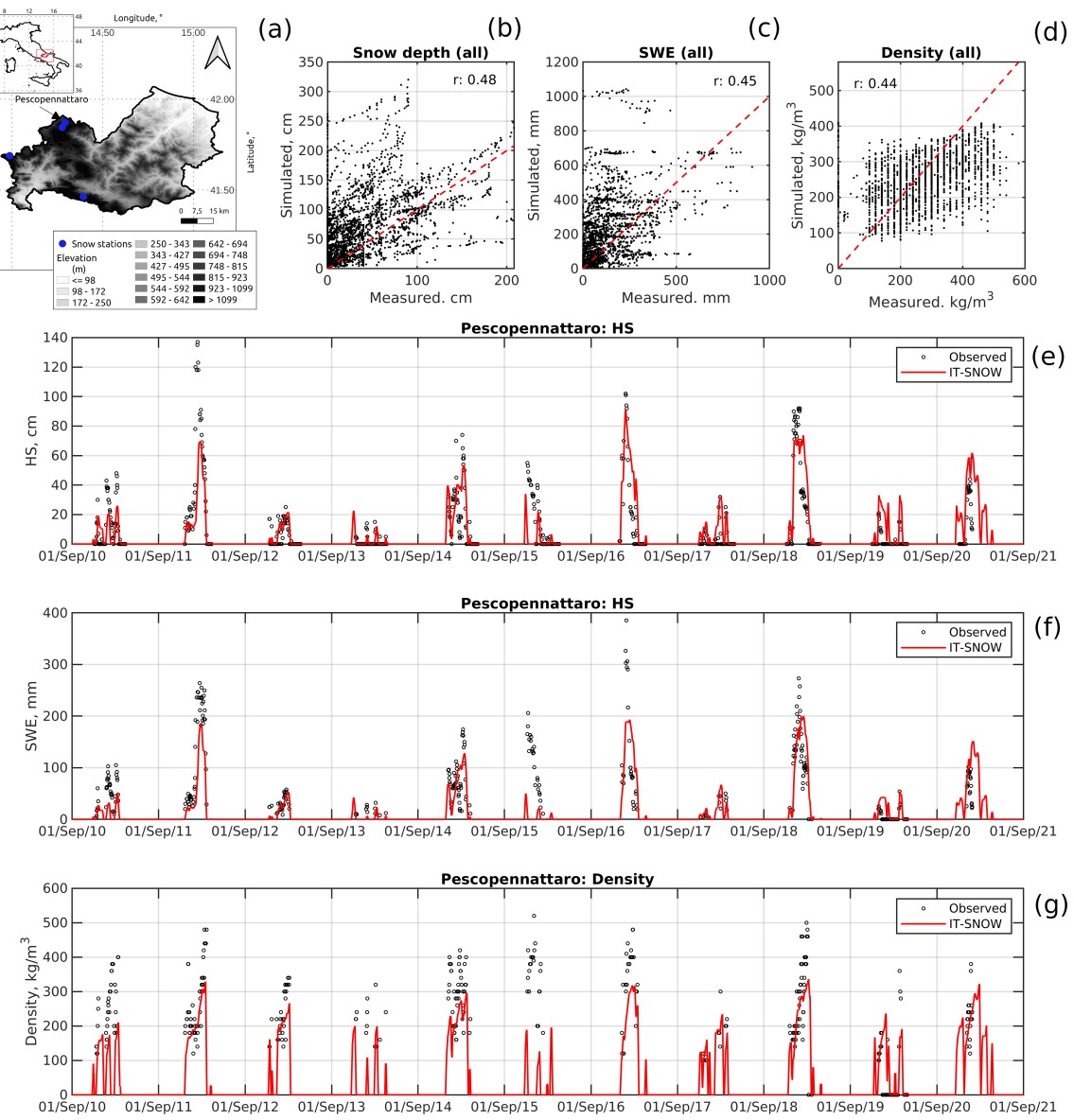

**Figure 8.** Evaluation of IT-SNOW in Molise. (a): topography of the focus region, along with sampling location of evaluation data; (b), (c), and (d): simulated vs. observed snow depth, SWE, and bulk-snow density at Molise snow stations, respectively; (e), (f), and (g): example of simulated vs. observed time-series of snow depth, SWE, and bulk-snow density at Pescopennattaro, respectively. Note that simulated snow depth, SWE, and density in panels (e), (f), and (g) was smoothed using a 5-day moving window for readibility. r is the Pearson's correlation coefficient, HS is snow depth, and SWE is Snow Water Equivalent. Density was measured with a resolution of $20 \, \text{kg}/\text{m}^3$, hence the discrete values of the point cloud along the x axis in panel d.

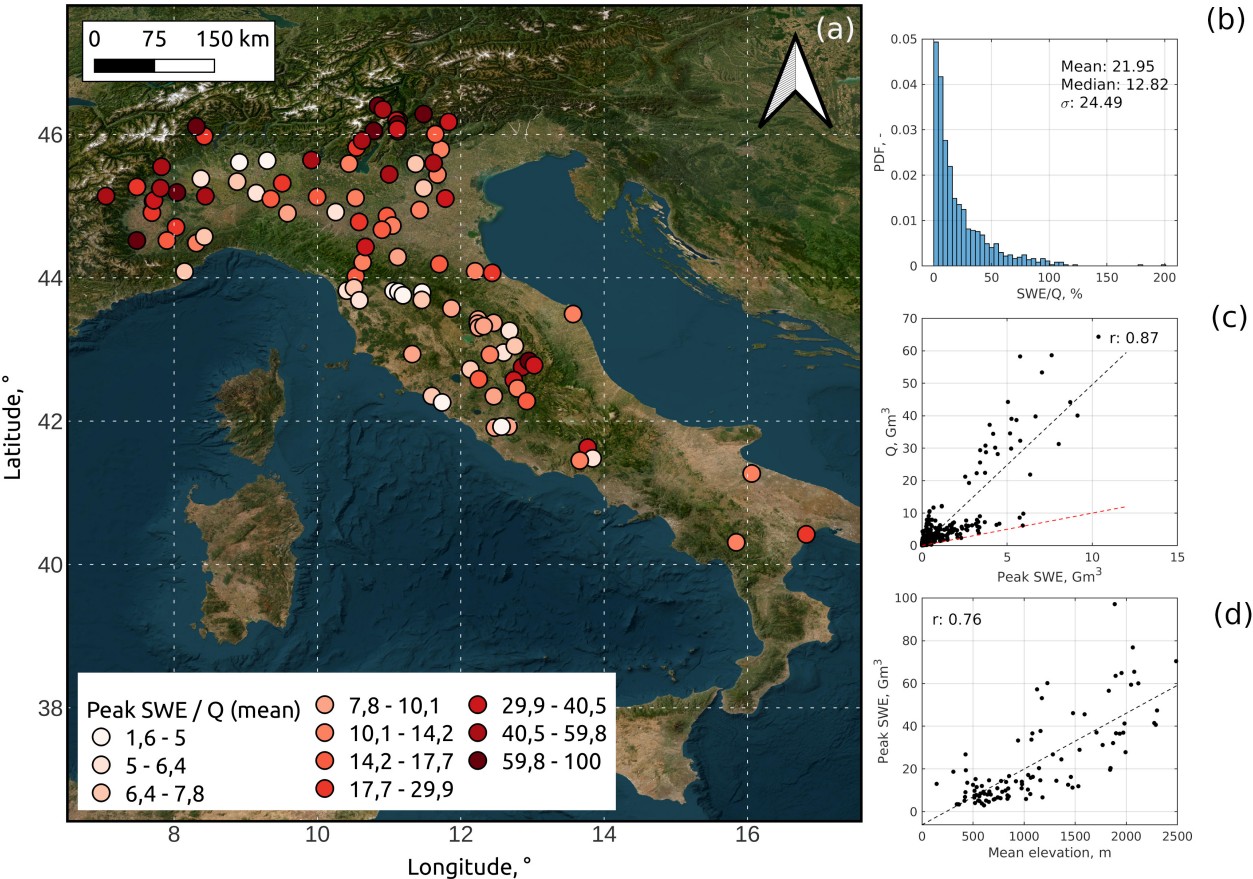

**Figure 9.** Indirect validation of IT-SNOW with streamflow data. (a): mean annual ratios (in %) between peak SWE and annual cumulative streamflow for a selection of 102 water basins with long-term, serially complete, and quality-checked time-series of streamflow; (b): frequency distribution of these annual ratios across all sections and years; (c): correlation between annual streamflow and annual peak SWE across all sections and years (red line is the 1:1 reference line, the black dashed line is a linear regression between peak SWE and annual streamflow); (d): correlation between mean peak SWE and river basin mean elevation (the black dashed line being a linear regression between these two data). Q is annual streamflow. Note that considered water sections of the central Po river valley do not account for snow accumulated in the Swiss Canton Ticino, which is not included in IT-SNOW (Canton Ticino represents about 5% of the Po river basin at its most downstream closure section). Background map: ESRI Satellite theme. SWE is expressed here in the form of total volume of water stored in snow, $Gm^3$, see Section 3.1.

handful of basins draining from central Apennines (Figure 9a). In these snow-dominated regions, the average ratio across the considered 9 water years locally exceeds 50 to 60%, especially in the high-elevation regions of north-western Italy and the Adige valley. Comparatively small watersheds on the central-western side of the Italian peninsula show significantly smaller

325 values, reflecting lower elevations and a more maritime and warmer climate.

As an additional, indirect validation of IT-SNOW, peak SWE is significantly correlated with annual streamflow (Pearson's correlation coefficient of 0.87, Figure 9c). In other words, peak SWE is a robust predictor of annual streamflow in Italy, which agrees with past experience in more snow-dominated regions of the world (Pagano et al., 2004; Rosenberg et al., 2011; Harrison and Bales, 2016). Again, as expected, peak SWE is smaller than annual streamflow (Figure 9c) and is significantly correlated with mean elevation of each watershed. Overall, these evaluations show consistency between IT-SNOW SWE estimates and the Italian water budget.

## 3.3 Sources of uncertainty

Like all reanalyses combining sparse data and a model over large domains, IT-SNOW is the result of a number of trade-off choices and epistemic uncertainties that should be taken into consideration while handling this dataset. The first source of uncertainty is represented by precipitation estimates, similar to any snow-hydrologic model simulation in mountain regions. The modified conditional merging approach used in IT-SNOW has already been extensively validated and shows robust performances for heavy precipitation events at national scale (relative error on high flows < 25% for 72% out of 241 Italian river sections when used to force a hydrologic model, see Bruno et al., 2021). However, it does not include explicit provisions for reconstructing precipitation orographic gradients (besides those captured by the location of stations and by radar images). Previous work in Aosta valley (Avanzi et al., 2021) and elsewhere (Lundquist et al., 2015; Zhang et al., 2017a; Avanzi et al., 2020), as well as results reported in this paper (Figure 5 and 7) show that including these orographic gradients is important to close the water budget of small, high-elevation, Alpine catchments. Targeted validations for such high-elevation, Alpine catchments will thus be the subject of future work.

A second source of uncertainty related to precipitation and more generally to in-situ weather data is data sparsity, and how this can affect spatialization in partially ungauged regions. While density of weather data in Italy is comparatively high (as order of magnitude, $\sim$1 station every 100 km$^2$ or more) and stations are routinely monitored and maintained by regional administrative authorities, we expect estimates for regions at the boundary of the Italian territory to show an inherently larger uncertainty than the rest of the country (because of a lack of input data outside the Italian territory). Data sparsity also limits the amount of snow-depth data that we can currently employ in assimilation, as well as the extent of evaluation regions in this paper (see Section 3). While estimating a real-time layer of uncertainty for this reanalysis product is currently non feasible, this will be target of future work.

Regarding the assimilation data, uncertainty in satellite SCA is in line with standards by the European Space Agency, as already discussed in Section 2. On the other hand, we noted that noise in snow-depth sensor data, along with the likely simplistic multilinear-regression approach used to spatialize snow depth across the landscape, often introduce artifacts in snow depth that translate into abrupt oscillations in snow depth values at the daily time scale (Figure 6). Another source of uncertainty in this regard is that we organized the Italian territory in 10 homogeneous regions, but no smoothing at the regional boundaries is currently in place. As a result, IT-SNOW may sometimes overestimate snow-depth variability across the boundaries of these homogeneous regions, and/or exaggerate the role of single topographic predictors of snow depth, such as aspect. An alternative

to our approach would be to directly assimilate remote-sensing products; ongoing efforts by the European Space Agency and others are moving towards this direction, and we aim to include new findings in this regard in our assimilation framework.

In terms of epistemic uncertainty, the S3M model relies on an enhanced temperature-index approach that was calibrated in Aosta valley and then extensively evaluated elsewhere in Italy (both in this paper and in other projects, such as Alfieri et al., 2022). Relying on the same parameters across the whole country could introduce some additional uncertainty at local scale. However, results in this paper show a credible reconstruction of melt dynamics even in areas with only occasional assimilation (Figure 8). In this regard, Bouamri et al. (2018) reported encouraging results when transferring model parameters of the same enhanced temperature-index approach to uncalibrated sites. A more influencing factor in this sense could stem from S3M not solving the full energy balance, but Magnusson et al. (2015) show that this is no key driver of model performance for snow bulk variables at daily scale.

## 4 Examples of use

By providing spatially explicit, high-resolution, and serially complete estimates of snow patterns, reanalyses like IT-SNOW have the potential to fill important knowledge gaps in hydrology, for example by elucidating the role of snow in supporting worldwide water security (Viviroli et al., 2007), or snow sensitivity to climate extremes like droughts (Hatchett and McEvoy, 2018). IT-SNOW also responds to pressing societal questions from operational water resources managers and decision makers, who need diversified information on snow distribution, amount, and interannual variability to make accurate decisions regarding water allocations and uses (Harrison and Bales, 2016). In this section, we show examples of how IT-SNOW can be used to answer two such societally relevant questions (Dozier, 2011; Margulis et al., 2015; Painter et al., 2016): (1) How much snow is accumulated across the landscape? (2) Where is it distributed?

### 4.1 How much snow is accumulated across the landscape?

Figure 10a shows total daily SWE across Italy for the whole period of record, again in the form of total water volume in snow (Gm$^3$, see Section 3.1). Peak SWE in Italy is on average $13.70 \pm 4.9$ Gm$^3$, with a minimum in 2017 ($\sim$5 Gm$^3$) and a maximum in 2014 ($\sim$23 Gm$^3$). SWE peaks on average on the $4^{th}$ of March $\pm 10$ days, that is, about one month earlier than the $1^{st}$-April reference date; this finding is in agreement with a recent reconsideration of this conventional date (Montoya et al., 2014). The earliest peak-SWE date was the $3^{rd}$ of February (2019), while the latest was the $26^{th}$ of March in 2013. Seasonal dynamics show signs of intraseasonal melt (e.g., see late 2011 to early 2012), which is expected in a Mediterranean region where cold-alpine and maritime snow types coexist (Sturm et al., 1995; Sturm and Liston, 2021). Nonetheless, snow seasons are temporally continuous, with no episode of intraseasonal melt out. Little to none carryover occurs between seasons.

A large fraction of Italian snow accumulates across the southern Alps, while snow accumulation on the Apennines is spatially more limited and – importantly – more variable from one season to the others (Figure 10b to l). This increased variability in the Apennines compared to the Alps agrees with sparse, but consistent previous work showing that snow in the Apennines – and particularly on their eastern side – is the result of intense, but highly seasonal lake-effect storms (Da Ronco et al., 2020). One

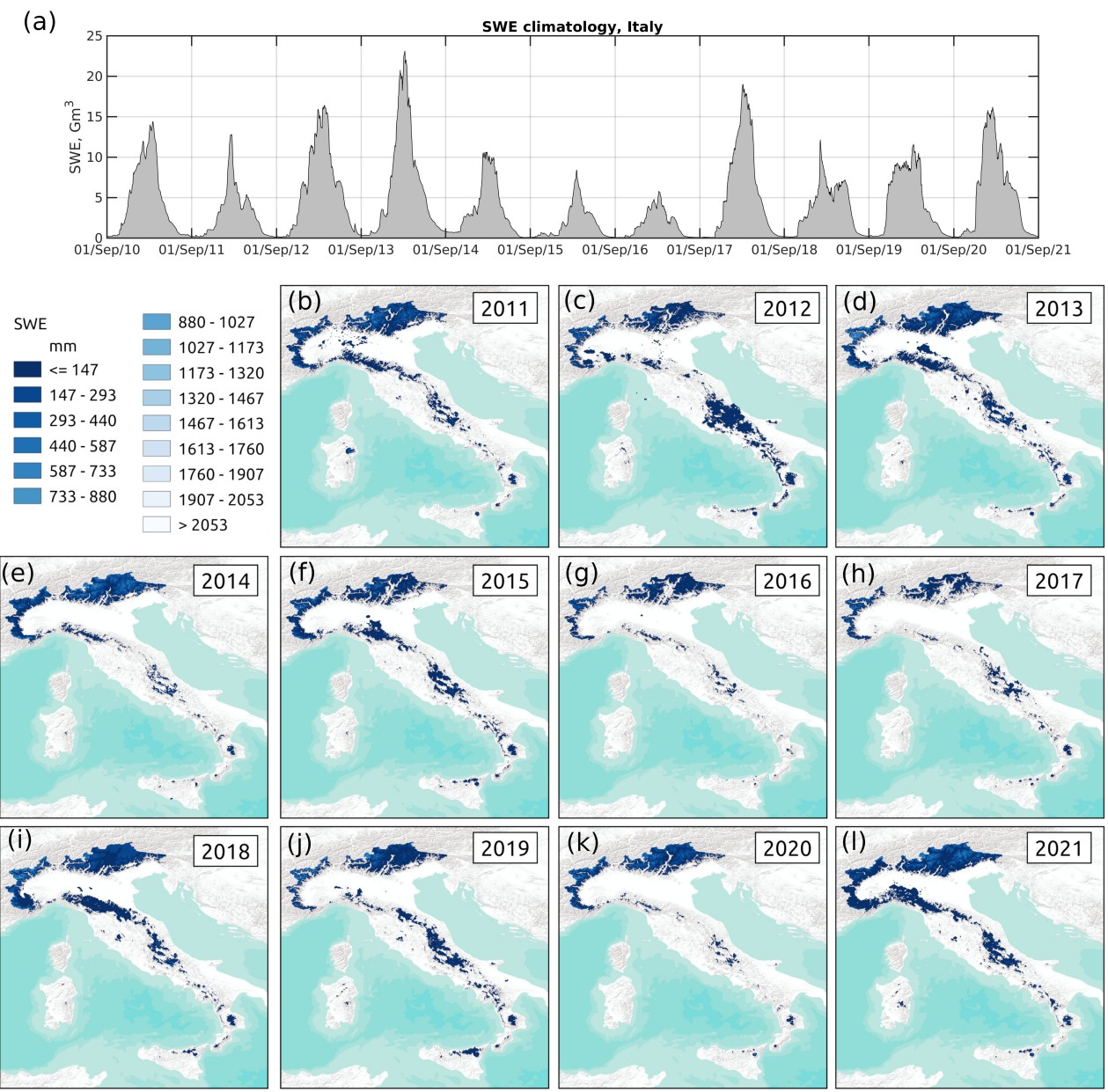

**Figure 10.** Example of use of IT-SNOW. (a): spatially aggregated total daily SWE across Italy; (b) to (l): mean annual SWE (November to June, water years 2011 to 2021). Background map: ESRI Terrain. SWE in panel (a) is expressed in the form of total volume of water stored in snow, Gm$^3$, see Section 3.1.

such event is evident in the 2012 reanalysis of IT-SNOW (Figure 10c), when central Italy was hit by extensive snowfalls as part of a continental cold wave (DemirtaÅ, 2017). Data for February 2012 report 150 cm or more of fresh snow at places, with

more precise estimates of return period for this event being challenged by the sparsity of the data network (Bisci et al., 2012). Another such exceptional event was the 2014 season in the southern Alps (Figure 10e), with 8 m of peak snow depth at 2000 m (200% of long-term mean, see Chiambretti et al., 2014). IT-SNOW also correctly captured rare, but significant low-elevation snowfall events (see water year 2011, Figure 10b).

The spatially and temporally consistent framework of IT-SNOW can aid not only climatological and hydrologic studies, for example by providing highly needed validation datasets of snow cover and SWE, but also operational water resources managers who routinely use SWE as an estimate of freshet volume (Harrison and Bales, 2016). Similar efforts have already been made in other regions like California, where Margulis et al. (2016) estimate about 20 $Gm^3$ of mean peak water volume in snow across the Sierra Nevada, or Japan, where Niwano et al. (2022) estimate 42.2 $Gm^3$ on average during the 2017-2022 winters. These estimate tally with that of IT-SNOW across the Italian mountain ranges (Figure 10), an area with a similar geographic span as the California Sierra Nevada or Japan but a less snow-dominated climate. With its high spatial resolution and fine temporal granularity, IT-SNOW can contribute to the quest for constraining the extent and volume of the world's cryosphere.

## 4.2    Where is it distributed?

Spatially aggregating mean winter SWE across major Italian watersheds shows that ∼52% of Italian snow-water resources accumulate across the Po river basin, the largest Italian water basin (Figure 11b). The second largest snow reservoir in Italy is expectedly the Adige river basin (23%), followed by the Piave, Tagliamento, and Brenta basins (6%, 3%, and 3%, respectively). Collectively, these five Alpine water basins host nearly 87% of Italian snow. A second hotspot of snow accumulation are central Apennines, with the Tevere river basin accumulating about 2% of national mean winter SWE. This and other three basins in central Italy (Aterno-Pescara, Garigliano, and Sangro) accumulate about 5% of national SWE, with the remaining 8-9% scattered across the remaining basins.

Median and quartiles of daily SWE for a selection of the most snow-dominated Italian basins show the typically seasonal dynamics of snow accumulation and melt, with peak-SWE date between water-year day ∼180 and ∼200 (that is, between the end of February and mid March, Figure 11c to i – SWE expressed as total volume of water stored in snow, $Gm^3$, see Section 3.1). As already noted in Figure 10, Apennines basins show much more interannual variability than Alpine basins, as evident in the larger spread between the first and third quartiles in Figure 11g to i compared to Figure 11c to f.

Italian basins also show significantly different melt rates: e.g., mean annual SWE for the Po river basin is 4 $Gm^3$, with an almost continuous snow cover and an almost symmetrical accumulation and melt season (Figure 11c). This symmetry significantly differs from more radiation-driven and maritime snow types, such as that of the Aterno-Pescara or Tagliamento rivers (Figure 11f and h), where the melt season is shorter than the accumulation season.

With its initial time span of 11 years, IT-SNOW-derived statistics like those from Figure 11c to Figure 11i can provide estimates of medium-term SWE variability and thus put accumulation at any given time (like those in Figure 10) into a broader context. Such a broader context is becoming all the more important in a warming and drier climate, particularly given the growing emergence of snow droughts as a specific typology of droughts (Harpold et al., 2017; Hatchett et al., 2017; Huning and AghaKouchak, 2020). In this regard, medium-term climatological bands like those in Figure 11 have already been used

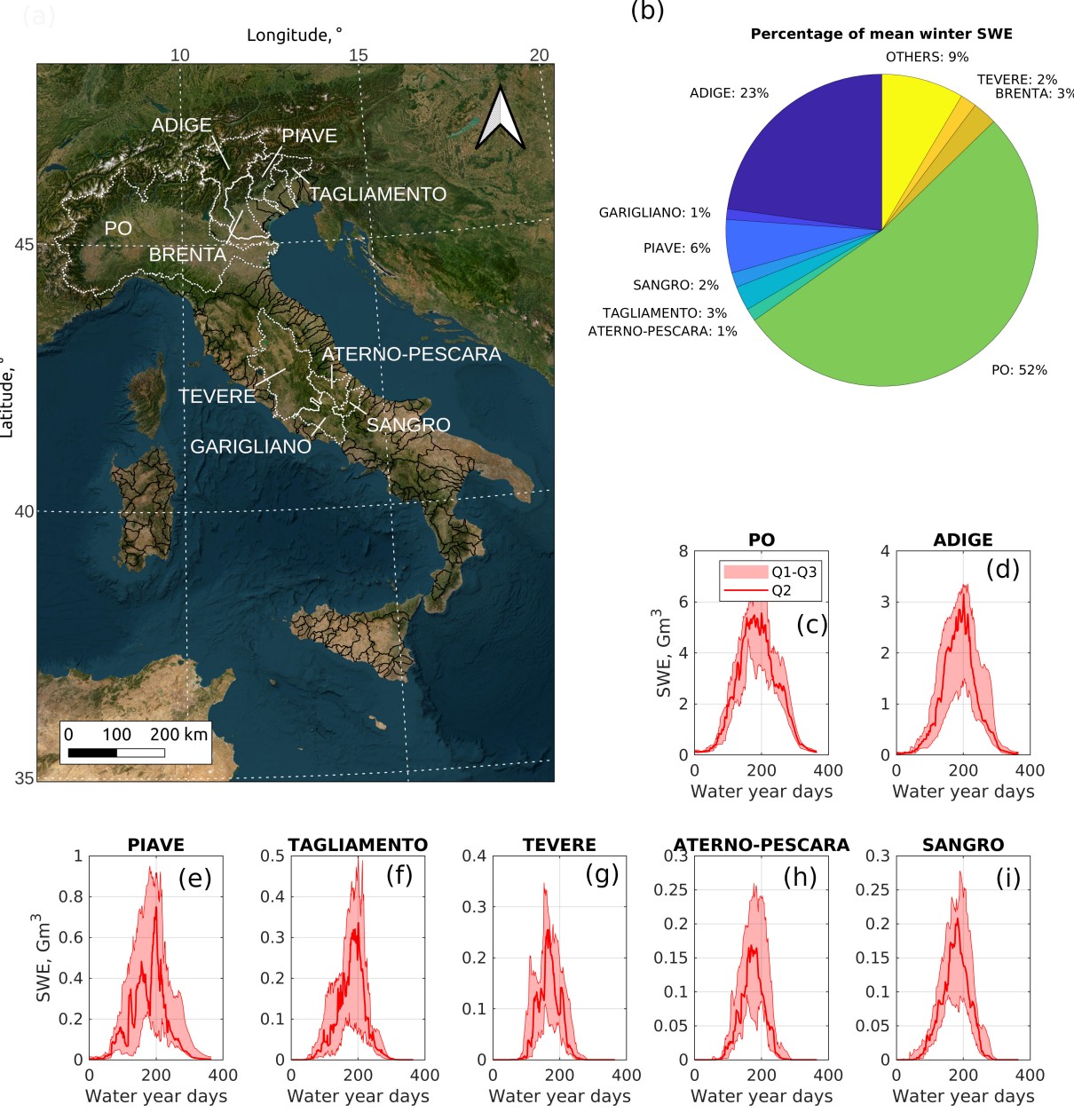

**Figure 11.** Example of use of IT-SNOW. (a) and (b): percentage of mean winter SWE that is accumulated across the major Italian river basins (right) and location of these major basins across Italy (left); (c) to (i): median and quartiles of daily basinwide SWE across a selection of the most snow-dominated basins in Italy. Q2 is the median, Q1 and Q3 are the first and third quartiles, respectively. Background map: ESRI Satellite theme. SWE is expressed here in the form of total volume of water stored in snow, $Gm^3$, see Section 3.1.

to contextualize the severe 2022 precipitation deficit in northern Italy, by showing that this deficit translated into ∼40% SWE compared to the 2009-2021 median (Toreti et al., 2022). While the initial time span of IT-SNOW snow is too short for rigorous deficit estimates, the operational chain delivering this reanalysis will provide yearly updates for future seasons, while an extension to the near past (say, 2002-2009) is also in consideration. Besides deficit analysis, we expect the notion of snow accumulation and melt rates to aid in other contexts, such as ecology (Slatyer et al., 2022) or climate-change assessments (Musselmann et al., 2017).

## 5   Data format

IT-SNOW is available in an open access framework at the following DOI: https://doi.org/10.5281/zenodo.7034956 (CC BY-NC 4.0, see Avanzi et al., 2022b). Data are organized in monthly netCDF files with zlib compression, each hosting a 3D matrix of daily maps for one variable of interest (SWE, snow depth, bulk snow density, and bulk liquid water content). Filename strings follow a consistent convention including variable name and month (e.g., ITSNOW_SWE_201009.nc for SWE data of September 2010. Variable labels and units are as follows: SWE for Snow Water Equivalent (mm w.e.), HS for Snow Depth (cm, Fierz et al., 2009), RhoS for bulk-snow density ($kg/m^3$), and ThetaW for bulk-liquid water content (vol%).

In addition to data, each netCDF file includes information regarding the reference system (variable crs, including well-known text strings for EPSG 4326), latitude and longitude matrices, and a time array. Compatibility with the NASA Panoply system (https://www.giss.nasa.gov/tools/panoply/, last access 23/08/2022) and with QGIS were both verified, including reprojection to a UTM metric system. Each netCDF file also includes metadata identifying contact points and curators.

## 6   Conclusions

We presented IT-SNOW, a spatially explicit and multi-year reanalysis of snow cover patterns across Italy at ∼500-m resolution. IT-SNOW is the reanalyzed output of S3M Italy, a cryospheric modeling chain operationally delivering spatial snapshots of snow water resources for civil-protection applications. Through S3M Italy, IT-SNOW ingests input data from thousands of automatic weather stations across the Italian territory, while assimilating daily snow-covered-area maps from ESA Sentinel-2, NASA MODIS, and EUMETSAT H-SAF products and multilinear regressions of on-the-ground snow-depth data. Validation results show little to no mean bias compared to C-SNOW, a state-of-the-art retrieval of snow depth from Sentinel 1, Root Mean Square Errors on the order of 30 to 60 cm for in-situ, measured snow depth and 90 to 300 mm for in-situ, measured Snow Water Equivalent, a strong (0.87) correlation between peak SWE and annual streamflow, and ratios between peak SWE and annual streamflow that are in line with expectations for this mixed rain-snow region (22% on average). Examples of use showed how IT-SNOW can both fill fundamental knowledge gaps in snow hydrology and support real-world applications, by answering recurring questions like "How much snow is accumulated across the Italian landscape?" (on average 13.70 ± 4.9 $Gm^3$ peak SWE), or "Where is it?" (∼52% and 21% across the Po and Adige river basins, respectively, with the remainder between less snow-dominated watersheds in north-eastern Italy and the central Apennines).

IT-SNOW will be annually updated and community engagement will be favored to maintain a high-resolution reanalysis of snow for Italy. Engagement will follow two lines of action: the first is institutional, as we are engaging with the Italian regional administrations to benchmark IT-SNOW against their long-standing systems providing local to regional estimates of SWE with methodologies that vary among the offices. The second is bottom-up, as we are designing tools to favor engagement through a GitHub discussion page hosted at https://github.com/c-hydro (last access 30/08/2022). The author team thus remains open to critical feedback from the user community on IT-SNOW accuracy, issues, and improvement opportunities.

## 7 Code and data availability

IT-SNOW is available in an open access framework at the following DOI: https://doi.org/10.5281/zenodo.7034956 (CC BY-NC 4.0, see Avanzi et al., 2022b).

Sources of data used are reported in the paper, and include the database of the Italian Regional Administrations and Autonomous Provinces, as accessible by CIMA Research Foundation through the Italian Civil Protection, the Aosta valley Environmental Protection Agency (snow courses), The Aosta valley Ufficio Neve e Valanghe (avalanche probing data), Meteomont (as available to the Molise Region, Molise snow data), the Lombardia Environmental Protection Agency (Lombardia data), and the C-SNOW initiative (Lievens et al., 2019).

The S3M model is open-source software. Official releases are available at https://zenodo.org/badge/latestdoi/350732564, while the following GitHub repository includes also minor updates: https://github.com/c-hydro/s3m-dev. The most important components of the S3M Italy operational chain are also open source, see https://github.com/c-hydro.

*Author contributions.* Francesco Avanzi, Simone Gabellani, and Fabio Delogu developed S3M Italy, with contributions from Silvia Puca, Alexander Toniazzo, Pietro Giordano, Marco Falzacappa, and Luca Ferraris. Francesco Avanzi, Simone Gabellani, Fabio Delogu, and Francesco Silvestro developed the S3M model, with evaluation contributions from Sara Ratto, Hervè Stevenin, Edoardo Cremonese, and Umberto Morra di Cella. Francesco Avanzi and Simone Gabellani validated IT-SNOW, with contributions from Lauro Rossi, Sara Ratto, Hervè Stevenin, Edoardo Cremonese, Umberto Morra di Cella, Antonio Cardillo, Matteo Fioletti, and Orietta Cazzulli. Flavio Pignone developed the historical dataset of precipitation according to the Modified Conditional Merging approach. Giulia Bruno processed streamflow data and assisted in the analysis of the streamflow-SWE relationship. Luca Pulvirenti, Giuseppe Squicciarino, and Elisabetta Fiori developed the operational chain providing daily Sentinel-2 snow-covered area maps and assisted with snow-covered area assimilation. Francesco Avanzi carried out the analyses and prepared the paper, with input from all coauthors.

*Competing interests.* The authors declare that they have no conflict of interest.

*Acknowledgements.* This research has been supported by the Italian Civil Protection Department.

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
