# Peer review of "IT-SNOW: a snow reanalysis for Italy blending modeling, in-situ data, and satellite observations (2010-2021)"

_Earth System Science Data, 2022_

## Author Comment (AC1)

**Reply to R1**

This paper employed an S3M model to blend multi-source in situ data and satellite observations to produce a spatially explicit and multi-year reanalysis of snow cover patterns across Italy at 500 m resolution. After validating with C-SNOW products, in situ measurements, and annual streamflow, this product has been proved effective, and could be potentially used in better understanding the contribution of snow on water resource management.

Despite of its significance, several issues still need to be resolved before a publication to ESSD. More detailed introduction about how to produce snow cover area from multi-source remote sensing images, and how to produce the reliable snow depth maps over the entire study could be sufficiently explained. In addition, it is suggested to add more indexes to validate the output snow estimates. Besides, the figures should be further refined so as to improve the overall presentation.

> We appreciated all these suggestions, which we will welcome in the revised version of the manuscript. Please see below for our detailed response and planned changes.

**Figure 1, the schematic of S3M was too simple, it is difficult to understand the key model/method, the data flow, and the output data.**

> Planned changes: We will rethink Figure 1 to better highlight key models and methods, as well as data flows.

**P115-116, please provide the elevation gradient for air temperature when you interpolated in situ air temperature.**

> Planned changes: We will provide summary statistics of lapse rates in the revised manuscript.

**P125-135, the snow covered area used in S3M model are produced from Sentinel 2, MODIS, and H-SAF initiatives. How did you produced snow cover area from Sentinel 2? How did you preprocess the MODIS and H-SAF data? Have you filled up the data gaps caused by cloud cover? How to fill the data gaps? How about the accuracy of the blended snow cover area products?**

> The workflow used in IT-SNOW is described at lines 126-139 and includes details on how SCA from Sentinel 2 was produced and its expected accuracy ("Sentinel-2 SCA is produced by operationally applying the Sen2Cor algorithm by ESA [...] and SCA maps were validated against snow depth sensors at national scale (not shown). Albeit lower in accuracy than snow-specific and high-resolution products like Theia (Gascoin et al., 2019a), Sen2Cor generally provides snow masks with accuracy above 80% (Main-Knorn et al., 2017)."). On the other hand, we did not report details on data sources and processing for MODIS and H-SAF products, for which we used the already available products as distributed by the respective data providers and did not execute any specific postprocessing. Besides mosaicking maps from multiple sources with different revisit times, no additional gap-filling for cloud coverage was performed (assimilation for cloud-covered pixels was thus foregone).
> Planned changes: We will add the above to the manuscript.

**Figure 2-4, and 8, please add scale bar and change the color of latitude and longitude grids from black to white or gray. It is difficult to identify detailed numerical value from current stretch effect of color bar.**

Planned changes: We agree and will modify maps as requested.

**P145-155, the in situ stations are primarily distributed in north areas in Figure 2 (a), so how did you produce the reliable snow depth maps over the entire study area? How about the overall accuracy of the daily snow depth maps over the entire 10 homogeneous regions? If some of the homogeneous regions are lack of snow depth data, how about the final output after running the S3M model for these regions?**

As we specified at lines 152ff, if less than 10 snow depth data points are available for a given homogeneous region, then spatialization and thus assimilation *for that homogeneous region* is foregone. Regarding accuracy, we then specified that "previous evaluations of this multilinear-regression model in Aosta valley show that it successfully captures orographic gradients in snow depth with an average uncertainty of $\pm 10\%$".

Planned changes: We will further clarify the above and will look for additional, independent measurements of snow depth across the Italian Alps to perform a specific evaluation of snow-depth maps.

**Figure 3, please add legend for (a); it is cannot see NaN class (in orange color) from (b); add scale bar for (a)-(c).**

Planned changes: Agreed, we will modify figures as requested.

**Figure 4, why did not show the results over the entire study area?**

Histograms in panel (c) and (d) do refer to the entire study region, while panel (a) and (b) refer to the two subregions of Italy with deep seasonal snow cover (see Figure 9).

Planned changes: We will specify the above in the caption of Figure 4, but would like to keep the figure as it is now in this regard.

**For the validation results, please also add Mean Absolute Error, Positive Mean Error, Negative Mean Error, and R Squared.**

Planned changes: Agreed, we will include these metrics as requested.

---

## Author Comment (AC2)

**Reply to R2**

This paper discussed a new spatially distributed Italian snow reanalysis through combining remote and in-situ measurement techniques with the already existing Snow Multidata Mapping and Modeling system (S3M). Evaluation of the reanalysis through comparison with separate in-situ (snow course) and remote sensing products (C-SNOW) showed reasonable error within the produced snow products including snow depth, snow water equivalent, and snow density. The output products showed agreement with inter- and intra-annual accumulation and ablation trends in various climatological regions throughout Italy where different snowpack characteristics exist.

The reanalysis and associated paper(s) are novel and show significant potential for use with climatological analysis and monitoring of the Italian snowpack, and the overall grammar and organization of the manuscript were good with minimal issues. However, revisions are required to improve the manuscript before it should be accepted to ESSD.

> We appreciated all these suggestions, which we will welcome in the revised version of the manuscript. Please see below for our detailed response and planned changes.

**It would be useful to have analysis of average error of snow depth, SWE, and density for each of the 10 homogeneous regions mentioned first on Line 146 and shown in Figure 3a. Given the distinct geographical and climatological characteristics of each region and non-uniform distribution of the in-situ sites, regional differences in error may be expected that would be important for users of this data to understand. It would also aid in the constraint of the relative importance of SWE in each of the basins discussed in Section 4.2 and Figure 10.**

> Changes to manuscript: We agree with this comment and we will provide this assessment. This is feasible using C-SNOW data, but we will also look for additional independent data points across the Italian Alps.

**Lines 115-116: Further information about these linear regressions should be presented. How were they derived and applied?**

> Once all temperature data for a given hour are downloaded, data are organized by meteo regions as dictated by the Italian Civil Protection (see the boundaries of these regions at https://mappe.protezionecivile.gov.it/it/mappe-rischi/bollettino-di-vigilanza, last access on November 3, 2022). For each of these regions, a region-specific hourly linear regression between air temperature and elevation is fitted, and then applied using the region's Digital Elevation Model to derive temperature maps.
> Changes to manuscript: Part of the information above was already reported in the manuscript, but we will integrate it with the passage about distributing using a Digital Elevation Model. Per comments by Reviewer 1, we will also provide a climatology of air temperature lapse rates.

**Line 162-163 and Figure 3: It is discussed that SCA maps are not assimilated but are used to clip pixels that are snow free from snow depth maps. Figure 3 shows the SCA and snow**

depth maps individually but it would be helpful to have an additional panel showing the post-SCA clipped snow depth map to highlight the data that is being assimilated.

Changes to manuscript: This is possible and will be done in the revised manuscript.

**Line 1: "The" at the beginning of the sentence can be omitted.**

Changes to manuscript: Agreed.

**Figure 1. Further detail is needed in this flowchart. Specific information on the meteorological variables as discussed on Line 101 as inputs should be displayed.**

Changes to manuscript: Agreed. This will be done in agreement with similar comments by Reviewer 1.

**Line 147-148: "expert knowledge". This doesn't need to be exhaustive, but it would be nice to know what other primary conditions were considered in the expert knowledge.**

Changes to manuscript: We will add such details. In this regard, we aggregated SOIUSA regions based on a tradeoff between maximizing data points availability in each region and complying with expected climatology (e.g., differentiating inner-Alpine valleys from coastal, maritime mountain ranges).

**Lines 233-237 and Figure 4d: Distribution of root mean squared error in Figure 4d shows a right skew. As such, median should be used instead of mean.**

Changes to manuscript: Agreed.

**Line 284 and Figure 8b: Same as above. Data shows right skew and median should be used rather than mean.**

Changes to manuscript: Agreed.

**Line 353: Should be '1st', not '1th'.**

Changes to manuscript: Agreed.

**Figure 3a: Add legend.**

Changes to manuscript: Agreed.

**Figures 2, 3, 4, 5, 6, 7, 9, and 10: Color blind-friendly color palettes should be implemented.**

Changes to manuscript: Agreed.

**Figures 2, 3, 4, and 8: Can't see lat/long grid lines. Suggest changing to more visible color.**

Changes to manuscript: Agreed.

**Figures 2, 3, 4 and 9: Increase size of color bars/scales and add additional values.**

Changes to manuscript: Agreed.

---

## Author Comment (AC3)

**Reply to R3**

This article presents a reanalysis of the snowpack conditions over the italian territory between 2011 and 2021. It uses a spatially distributed snowpack model (S3M) forced with gridded in-situ observations from automatic weather stations (AWS) and radar. The simulated snow depth is corrected by the assimilation of snow depth measured at AWS gridded thanks to a multilinear regression model and adjusted by satellite-based snow cover maps. The uncertainty of the reanalysis is estimated with Sentinel-1 derived snow depth (C-SNOW) and in-situ snow depth and SWE measurements.

This high-quality reanalysis will sure be useful for many applications. The article is well-written and seems comprehensive, covering most aspects of this work. The methods and results are well presented. I believe that the following points should be addressed by the authors before publication. Below are smaller suggestions and details to help improve the article.

> We appreciated all these suggestions, which we will welcome in the revised version of the manuscript. Please see below for our detailed response and planned changes.

The figures need improvement, especially figures 2 and 3. Each map needs a title. The axes should be labeled, a scale added. The colorbar choice often does not allow a clear reading of the maps. The colorbar legend is often too small and with too few labels. See detailed comments on each figure below.

> Changes to manuscript: Agreed, we will welcome all these recommendations.

Some data and methods information seems missing. I could not find which digital elevation model is used (what source, what resolution) or if the land cover is taken into account. It would be good to mention if the interaction between the snowpack and the vegetation, such as the forests, are considered.

> We used an Italian Digital Elevation Model as made available by The Italian Institute for Environmental Protection and Research, ISPRA. The Digital Elevation Model was originally at 20 m, which we resampled at 200 m using an averaging method. Land cover or snow-forest interactions are not taken into account by S3M Italy.
> Changes to manuscript: Agreed, we will include all points above in the revised manuscript.

Other reanalysis over the swiss, the austrian and the french Alps (Fiddes et al., 2019, Olefs et al., 2020, Vernay et al., 2021) are mentionned. Although the methods are largely different in each work, it would be interesting to compare the uncertainty of these works.

> Changes to manuscript: We will add a Discussion on this in the manuscript.

L2 "+" disturbing notation. I suggest using "over", ">" or just give the exact value. To be homogeneized in the text.

> Changes to manuscript: Agreed.

**L9 "no mean bias" rather than "none"? (L421 as well)**

> Changes to manuscript: Agreed.

**L14 If ever the variability of the peak SWE date is available, it could be interesting to provide it.**

> Changes to manuscript: Yes, this is available and will be included in the manuscript.

**L25 "(Serreze et al., 1999; Skiles et al., 2018)" you might want to cite Li et al. (2017) in which the contribution of the snowpack to the runoff is indeed calculated. It seems like Serreze et al. (1999) only compared the solid precipitation amount to the total runoff and Skiles et al. (2018) cites Bardsley et al. (2013) for the 80% number.**

> Changes to manuscript: We totally agree and will amend the manuscript as suggested.

**L38 "lidar" in Deems et al. (2013), "Lidar measurement of snow depth : a review". To correct everywhere.**

> Changes to manuscript: Agreed.

**L38 "airborne lidar"? otherwise the list mixes methods (lidar, optic) and plateform (drone, satellite).**

> Changes to manuscript: Agreed.

**L47 and further in the text: what is a "dynamic model"?**

> We used the term "dynamic model" to indicate digital replica of environmental systems, in this case snow and glaciers. To our knowledge, this complies with general jargon in the hydrologic literature.
> Changes to manuscript: We will replace "dynamic model" with "model", which is more general and less ambiguous.

**L67 GlobSnow: maybe worth to mention that it is not available in mountain areas?**

> Changes to manuscript: Agreed.

**L91. A bit confusing with S3M, S3M Italy and IT-SNOW. Maybe add "the reanalysis IT-SNOW"**

> Changes to manuscript: Agreed.

**L100 "**" => I was disturbed by this notation without letters. Maybe use "hh" instead?**

> Changes to manuscript: Agreed.

**L100 Maybe precise the period covered by the inputs: is it only of the last hour?**

> We generate inputs every hour, since S3M Italy has a hourly time step. For redundancy

reasons, and to fill potential gaps due to occasional malfunctioning and/or failures, every hour automatic procedures check the existence of inputs for the last 30 hours.

Changes to manuscript: We will add the above to the manuscript.

**L108 RMSE of 1 mm, please provide the typical precipitation observed.**

Changes to manuscript: Agreed.

**L112 "spatialized" at what resolution?**

Changes to manuscript: We will add resolutions in the main text.

**L115 It would be very useful to provide the distribution of the temperature lapse-rate, even if supplement in necessary. This study from Navaro-Serrano et al. (2018) might help if you need to compare your temperature lapse-rate to similar regions.**

Changes to manuscript: Agreed, as also recommended by reviewer 1 and 2.

**L118 I would suggest rewording along "An unique estimate of the precision of these data is not available as the type of sensor installed varies from one region to another. The installation and the maintenance of the sensors..."**

Changes to manuscript: Agreed.

**L122 "remapped" quite vague. Cropped?**

Changes to manuscript: Agreed.

**L124 "each region to tailor" unclear. What is the exact meaning of "region" here? What is tailoring S3M?**

In Italy, each administrative region (first-level administrative divisions of the Italian Republic) has civil-protection offices that have access to real-time outputs of S3M Italy. Each of these administrative regions may potentially ask for region-specific parametrizations.

Changes to manuscript: While the above was the intended meaning of *tailoring* in this context, we will remove that passage as it is superfluous information.

**L128 "Sentinel-2"**

Changes to manuscript: Agreed.

**L129 How do you manage the overlapping images? Putting on top the most recent?**

Changes to manuscript: Yes, exactly. We will specify this in the text.

**L137. "Not shown". Could be added in supplement maybe?**

Changes to manuscript: Agreed, we will provide such statistics as a supplement.

**L146 Please provide the number of snow depth sensor.**

Changes to manuscript: Agreed.

**L151 "remapped"? unclear.**

Changes to manuscript: Same as above, we will use "cropped" as suggested earlier.

**L159. "For each time instant" not clear. Could be deleted.**

Changes to manuscript: Agreed, this will be deleted.

**L163 What happends if snow in SCA observation but not in S3M? "preserving" is a bit unclear, maybe use "leaving without snow..."?**

Changes to manuscript: Correct, we will revise as suggested.

**L170 "The duration"?**

Changes to manuscript: Agreed.

**L171 "1.3 h" give it in h and min.**

Changes to manuscript: Agreed.

**L172 "AM" a.m.?**

Changes to manuscript: Agreed.

**L184 Given the resolution, it seems like at least the last "57" can be dropped.**

Changes to manuscript: Agreed.

**L194 Some precisions about C-SNOW product would be welcome. First, it is only available for dry snow, that is accumulation period, isn't it? Second, some part of Italy seems not covered by C-SNOW (grey area in Fig. 1 of Lievens et al., 2019). Finally, is C-SNOW completely independant from IT-SNOW? C-SNOW was calibrated on snow depth from AWS.**

Correct, C-SNOW is only available for dry snow. It is also correct that only mountain regions are covered by the product. Finally, the reviewer is right that some (but not all) snow-depth data from Italian AWS were used in developing C-SNOW, along with a variety of other snow-depth data from the Northern Hemisphere. In any case, C-SNOW remains the only, high-resolution and temporally dense remote-sensing product of snow depth that can be used to validate IT-SNOW across the Italian mountain ranges.
Changes to manuscript: We will add the above to the manuscript.

**L211 "ASL" a.s.l.?**

Changes to manuscript: Agreed.

**L232 Is it not possible to make it "3.2 Results" and then sub-sections (3.2.n) for the different**

**data sources?**

> Changes to manuscript: Agreed.

**L235 how did you compute the RMSE? Between time-series at each pixel? Please write the MAD from Lievens et al. (2019).**

> Yes, we computed RMSE between time-series at each pixel.
> Changes to manuscript: We will add the above and the requested MAD to the manuscript.

**L243 Please provide values for the bias. A table summing all the statistics evaluation would be really helpful.**

> Changes to manuscript: Agreed.

**L248 Then there is little information brought by the comparison of IT-SNOW density with station density since the density is derived from snow depth and SWE and snow depth and SWE are also compared to IT-SNOW.**

> This is correct, but at the same time we think it is important to provide error estimates for density given that this is one of the four data layers provided by IT-SNOW.
> Changes to manuscript: No change.

**L253 Please provide bias values.**

> Changes to manuscript: Agreed.

**L284 "102 gauge stations"?**

> Changes to manuscript: Agreed.

**L294 "Again,..."**

> Changes to manuscript: Agreed.

**L306 "evalution results"=> "the results"**

> Changes to manuscript: Agreed.

**L307-309 Cut this long sentence int two.**

> Changes to manuscript: Agreed.

**L313 "peripheral"? geographicaly peripheral? The Alps are on the periphery of Italy but the station density is high...not clear.**

> We meant areas that are at the periphery of the Italian territory. While station density in these regions may remain locally high, spatially distributed information like blended gauge-radar precipitation will have an inherently larger uncertainty (because of a lack of, e.g., available radar information outside the Italian territory).

Changes to manuscript: We will elaborate on the above in the manuscript.

**L317 not clear if talking about the SCA of the Sentinel-2/MODIS product or from IT-SNOW.**

Changes to manuscript: We meant SCA from satellites. This will be clarified.

**L331-334 sentence is too long. To be cut.**

Changes to manuscript: Agreed.

**L334 "apriori appear"? please rephrase.**

Changes to manuscript: We will delete "a-priori" here as it is confusing.

**L336 "Thus, quantifying this uncertainty in still elusive at this stage." I dont understand where this conclusion stems from.**

This conclusion stems from previous work, such as Magnusson et al. (2015) or Avanzi et al. (2016), showing that solving the energy balance is no key driver of model performance for snow bulk variables at daily scale.
Changes to manuscript: We will further clarify this point.

**L336: "in"=>is**

Changes to manuscript: Agreed.

**L337 To move earlier in the description of the data or method.**

Changes to manuscript: Agreed.

**L344 "basic science"? Please reformulate.**

Changes to manuscript: Agreed.

**L348 I like the catchy quesions. However, "what is it doing?" is not so clear and it does not appear in the conclusion. Could it be deleted? I also suggest more detailed formulation "How much is accumulated in total? Where/how is it spatially distributed?"**

Changes to manuscript: Agreed.

**L353 1st**

Changes to manuscript: Agreed.

**L353 "(this finding is in agreement with a recent reconsideration of this conventional date, see Montoya et al., 2014)" get this sentence out of the brackets**

Changes to manuscript: Agreed.

**L364 "anecdotal data" unclear if they are data from IT-SNOW or non-scientific data.**

>   These data come from newspapers and/or other publications, as reported in the main text.
>   Changes to manuscript: We will revise text here and remove the word "anecdotal".

**L366 "150+ cm of fresh snow in 24 hours" give the date of this event.**

>   Changes to manuscript: Agreed.

**Figure 2. Provide title in the figure for each subplot, next to (a,b,c). Provide more values for the colorbar of b and c. In a, is there no station with several type of measurements? Or are they hidden because points overlapp?**

>   Changes to manuscript: Agreed. We will also specify that some stations may host several types of measurements.

**Make b colorbar symetric so that 0° is linked to the yellow color. In b, keep the same precision (14,8//-16,77) and meaningful values (-10,0,10. . . ).**

>   Changes to manuscript: Agreed.

**In the legend: "by S3M Italy and thus the IT-SNOW reanalysis" => "by S3M Italy to produce IT-SNOW." Confusing otherwise.**

>   Changes to manuscript: Agreed.

**Scale is missing as well as xlabel and ylabel.**

>   Changes to manuscript: Agreed.

**If you find a way to make all subplots fits in only one line, it would make better use of the space.**

>   Changes to manuscript: Agreed.

**Figure 3 See comments for Figure 2 that can be applied here. For b, why is there a transparent area without data which is not of the colour of the NaN provided in the legend.**

>   Changes to manuscript: Agreed, we will revise the figure and the blank vs. NaN issue.

**Figure 4 This figure is much more readable. Add xlabel, ylabel to the maps and make sure color scale is symetric centered on 0. Improve the colorbar (see above).**

>   Changes to manuscript: Agreed.

**Figure 5 Suggestion for future figures: b and c would be better plotted with a heat-map or at least some transparency of the points.**

>   Changes to manuscript: Agreed.

**Figure 8 Make the color scale continuous, it is really hard to read the map otherwise.**

    Changes to manuscript: Agreed.

**References**

Avanzi, F., De Michele, C., Morin, S., Carmagnola, C.M., Ghezzi, A., Lejeune, Y., 2016. Model complexity and data requirements in snow hydrology: seeking a balance in practical applications. Hydrological Processes .

Magnusson, J., Wever, N., Essery, R., an A. Winstral, N.H., Jonas, T., 2015. Evaluating snow models with varying process representations for hydrological applications. Water Resources Research 51, 2707 – 2723.

---

## Author Response (AR1)

Savona (Italy)

December 22, 2022

Dear Editors,

We would like to re-submit the manuscript *IT-SNOW: a snow reanalysis for Italy blending modeling, in-situ data, and satellite observations (2010-2021)* to ESSD.

We have extensively revised the manuscript based on comments from the three reviewers and would like to thank all of you for finding the time to review our manuscript. We confirm that all requested changes were feasible and we welcomed all of them at our best.

Please find attached our point-by-point replies and the new version of our manuscript for details. We also attached a version of the manuscript with tracked changes.

With our best regards,

*Francesco Avanzi and coauthors*

**Reply to R1**

This paper employed an S3M model to blend multi-source in situ data and satellite observations to produce a spatially explicit and multi-year reanalysis of snow cover patterns across Italy at 500 m resolution. After validating with C-SNOW products, in situ measurements, and annual streamflow, this product has been proved effective, and could be potentially used in better understanding the contribution of snow on water resource management.

Despite of its significance, several issues still need to be resolved before a publication to ESSD. More detailed introduction about how to produce snow cover area from multi-source remote sensing images, and how to produce the reliable snow depth maps over the entire study could be sufficiently explained. In addition, it is suggested to add more indexes to validate the output snow estimates. Besides, the figures should be further refined so as to improve the overall presentation.

>We appreciated all these suggestions, which we welcomed in the revised version of the manuscript. Please see below for our detailed response and Changes.

**Figure 1, the schematic of S3M was too simple, it is difficult to understand the key model/method, the data flow, and the output data.**

>Changes: We revised Figure 1 to better highlight key models and methods, as well as data flows.

**P115-116, please provide the elevation gradient for air temperature when you interpolated in situ air temperature.**

>Changes: We provided summary statistics of lapse rates in the revised manuscript (see lines **123ff** and Figure 3).

**P125-135, the snow covered area used in S3M model are produced from Sentinel 2, MODIS, and H-SAF initiatives. How did you produced snow cover area from Sentinel 2? How did you preprocess the MODIS and H-SAF data? Have you filled up the data gaps caused by cloud cover? How to fill the data gaps? How about the accuracy of the blended snow cover area products?**

>The workflow used in IT-SNOW was described at lines 126-139 (previous version of the manuscript), including details on how SCA from Sentinel 2 was produced and its expected accuracy ("Sentinel-2 SCA is produced by operationally applying the Sen2Cor algorithm by ESA [...] and SCA maps were validated against snow depth sensors at national scale (not shown). Albeit lower in accuracy than snow-specific and high-resolution products like Theia (Gascoin et al., 2019a), Sen2Cor generally provides snow masks with accuracy above 80% (Main-Knorn et al., 2017)."). On the other hand, the previous version of the manuscript did not report details on data sources and processing for MODIS and H-SAF products, for which we used the already available products as distributed by the respective data providers and did not execute any specific postprocessing. Besides mosaicking maps from multiple sources with different revisit times, no additional gap-filling for cloud coverage was performed (assimilation

for cloud-covered pixels was thus foregone).

     Changes: We added the above to the manuscript (see lines **135ff**).

**Figure 2-4, and 8, please add scale bar and change the color of latitude and longitude grids from black to white or gray. It is difficult to identify detailed numerical value from current stretch effect of color bar.**

     Changes: We modified maps as requested (see current Figures 2, 4, 5, and 9).

**P145-155, the in situ stations are primarily distributed in north areas in Figure 2 (a), so how did you produce the reliable snow depth maps over the entire study area? How about the overall accuracy of the daily snow depth maps over the entire 10 homogeneous regions? If some of the homogeneous regions are lack of snow depth data, how about the final output after running the S3M model for these regions?**

     As we specified at lines 152ff in the previous version of the manuscript, if less than 10 snow depth data points are available for a given homogeneous region, then spatialization and thus assimilation *for that homogeneous region* is foregone. Regarding accuracy, we then specified that "previous evaluations of this multilinear-regression model in Aosta valley show that it successfully captures orographic gradients in snow depth with an average uncertainty of ±10%".

     Changes: We clarified the above and added examples of biases and Root Mean Square Errors of snow-depth maps against in-situ and remote-sensing data in Aosta valley and at national scale (see lines **159ff**).

**Figure 3, please add legend for (a); it is cannot see NaN class (in orange color) from (b); add scale bar for (a)-(c).**

     Changes: Modified as requested (see current Figure 4).

**Figure 4, why did not show the results over the entire study area?**

     Histograms in panel (c) and (d) do refer to the entire study region, while panel (a) and (b) refer to the two subregions of Italy with deep seasonal snow cover (see Figure 9).

     Changes: We specified the above in the caption of now Figure 5.

**For the validation results, please also add Mean Absolute Error, Positive Mean Error, Negative Mean Error, and R Squared.**

     Changes: Agreed and included as requested (see Table 1 and lines **239ff**).

**Reply to R2**

This paper discussed a new spatially distributed Italian snow reanalysis through combining remote and in-situ measurement techniques with the already existing Snow Multidata Mapping and Modeling system (S3M). Evaluation of the reanalysis through comparison with separate in-situ (snow course) and remote sensing products (C-SNOW) showed reasonable error within the produced snow products including snow depth, snow water equivalent, and snow density. The output products showed agreement with inter- and intra-annual accumulation and ablation trends in various climatological regions throughout Italy where different snowpack characteristics exist.

The reanalysis and associated paper(s) are novel and show significant potential for use with climatological analysis and monitoring of the Italian snowpack, and the overall grammar and organization of the manuscript were good with minimal issues. However, revisions are required to improve the manuscript before it should be accepted to ESSD.

> We appreciated all these suggestions, which we welcomed in the revised version of the manuscript. Please see below for our detailed response and planned changes.

It would be useful to have analysis of average error of snow depth, SWE, and density for each of the 10 homogeneous regions mentioned first on Line 146 and shown in Figure 3a. Given the distinct geographical and climatological characteristics of each region and non-uniform distribution of the in-situ sites, regional differences in error may be expected that would be important for users of this data to understand. It would also aid in the constraint of the relative importance of SWE in each of the basins discussed in Section 4.2 and Figure 10.

> Changes: We added this assessment (see lines **266ff** and Figures S2 and S3 in the Supplement). This assessment was carried out using C-SNOW data, as they cover the whole area of interest.

**Lines 115-116: Further information about these linear regressions should be presented. How were they derived and applied?**

> Once all temperature data for a given hour are downloaded, data are organized by meteo regions as dictated by the Italian Civil Protection (see the boundaries of these regions at https://mappe.protezionecivile.gov.it/it/mappe-rischi/bollettino-di-vigilanza, last access on November 3, 2022). For each of these regions, a region-specific hourly linear regression between air temperature and elevation is fitted, and then applied using the region's Digital Elevation Model to derive temperature maps.
>
> Changes: Part of the information above was already reported in the manuscript, but we integrated it with the passage about distributing using a Digital Elevation Model (see lines **164ff**). Per comments by Reviewer 1, we provided a climatology of air temperature lapse rates (see lines **123ff** and Figure 3).

**Line 162-163 and Figure 3: It is discussed that SCA maps are not assimilated but are used to clip pixels that are snow free from snow depth maps. Figure 3 shows the SCA and snow**

depth maps individually but it would be helpful to have an additional panel showing the post-SCA clipped snow depth map to highlight the data that is being assimilated.

> Changes: Done (see now Figure 4).

**Line 1: "The" at the beginning of the sentence can be omitted.**

> Changes: Done.

**Figure 1. Further detail is needed in this flowchart. Specific information on the meteorological variables as discussed on Line 101 as inputs should be displayed.**

> Changes: Done, including similar comments by Reviewer 1.

**Line 147-148: "expert knowledge". This doesn't need to be exhaustive, but it would be nice to know what other primary conditions were considered in the expert knowledge.**

> Changes: Details added (see lines **159ff**).

**Lines 233-237 and Figure 4d: Distribution of root mean squared error in Figure 4d shows a right skew. As such, median should be used instead of mean.**

> Changes: Done (see lines **256ff**).

**Line 284 and Figure 8b: Same as above. Data shows right skew and median should be used rather than mean.**

> Changes: Done (see lines **218ff**).

**Line 353: Should be '1st', not '1th'.**

> Changes: Done.

**Figure 3a: Add legend.**

> Changes: Done (see now Figure 4).

**Figures 2, 3, 4, 5, 6, 7, 9, and 10: Color blind-friendly color palettes should be implemented.**

> Changes: We did our best to comply with this important comment.

**Figures 2, 3, 4, and 8: Can't see lat/long grid lines. Suggest changing to more visible color.**

> Changes: Done.

**Figures 2, 3, 4 and 9: Increase size of color bars/scales and add additional values.**

> Changes: Done.

**Reply to R3**

This article presents a reanalysis of the snowpack conditions over the italian territory between 2011 and 2021. It uses a spatially distributed snowpack model (S3M) forced with gridded in-situ observations from automatic weather stations (AWS) and radar. The simulated snow depth is corrected by the assimilation of snow depth measured at AWS gridded thanks to a multilinear regression model and adjusted by satellite-based snow cover maps. The uncertainty of the reanalysis is estimated with Sentinel-1 derived snow depth (C-SNOW) and in-situ snow depth and SWE measurements.

This high-quality reanalysis will sure be useful for many applications. The article is well-written and seems comprehensive, covering most aspects of this work. The methods and results are well presented. I believe that the following points should be addressed by the authors before publication. Below are smaller suggestions and details to help improve the article.

> We appreciated all these suggestions, which we welcomed in the revised version of the manuscript. Please see below for our detailed response and planned changes.

The figures need improvement, especially figures 2 and 3. Each map needs a title. The axes should be labeled, a scale added. The colorbar choice often does not allow a clear reading of the maps. The colorbar legend is often too small and with too few labels. See detailed comments on each figure below.

> Changes: We welcomed all these recommendations at our best, especially with regard to maps in former Figures 2 and 3.

Some data and methods information seems missing. I could not find which digital elevation model is used (what source, what resolution) or if the land cover is taken into account. It would be good to mention if the interaction between the snowpack and the vegetation, such as the forests, are considered.

> We used an Italian Digital Elevation Model as made available by The Italian Institute for Environmental Protection and Research, ISPRA. The Digital Elevation Model was originally at 20 m, which we resampled at 200 m using an averaging method. Land cover or snow-forest interactions are not taken into account by S3M Italy.
> Changes: We included all points above in the revised manuscript (see lines **129ff** and **95ff**).

Other reanalysis over the swiss, the austrian and the french Alps (Fiddes et al., 2019, Olefs et al., 2020, Vernay et al., 2021) are mentionned. Although the methods are largely different in each work, it would be interesting to compare the uncertainty of these works.

> Changes: We added some Discussion on this in the manuscript (see lines **299ff**).

L2 "+" disturbing notation. I suggest using "over", ">" or just give the exact value. To be homogeneized in the text.

       Changes: Done throughout the text (e.g, see line **2**).

**L9 "no mean bias" rather than "none"? (L421 as well)**

       Changes: Done (e.g., see line **10**).

**L14 If ever the variability of the peak SWE date is available, it could be interesting to provide it.**

       Changes: Included in the manuscript (e.g., see line **14**).

**L25 "(Serreze et al., 1999; Skiles et al., 2018)" you might want to cite Li et al. (2017) in which the contribution of the snowpack to the runoff is indeed calculated. It seems like Serreze et al. (1999) only compared the solid precipitation amount to the total runoff and Skiles et al. (2018) cites Bardsley et al. (2013) for the 80% number.**

       Changes: Done (see line **25**).

**L38 "lidar" in Deems et al. (2013), "Lidar measurement of snow depth : a review". To correct everywhere.**

       Changes: Done (see line **37**).

**L38 "airborne lidar"? otherwise the list mixes methods (lidar, optic) and plateform (drone, satellite).**

       Changes: Done (see line **37**).

**L47 and further in the text: what is a "dynamic model"?**

       We used the term "dynamic model" to indicate digital replica of environmental systems, in this case snow and glaciers. To our knowledge, this complies with general jargon in the hydrologic literature.

       Changes: We replaced "dynamic model" with "model", which is more general and less ambiguous.

**L67 GlobSnow: maybe worth to mention that it is not available in mountain areas?**

       Changes: Done (see line **66**).

**L91. A bit confusing with S3M, S3M Italy and IT-SNOW. Maybe add "the reanalysis IT-SNOW"**

       Changes: Done (see line **91**).

**L100 "**" => I was disturbed by this notation without letters. Maybe use "hh" instead?**

       Changes: Done (see line **101**).

**L100 Maybe precise the period covered by the inputs: is it only of the last hour?**

We generate inputs every hour, since S3M Italy has a hourly time step. For redundancy reasons, and to fill potential gaps due to occasional malfunctioning and/or failures, every hour automatic procedures check the existence of inputs for the last 30 hours.

Changes: We added the above to the manuscript (see lines **103ff**).

**L108 RMSE of 1 mm, please provide the typical precipitation observed.**

Changes: Done (see line **114ff**).

**L112 "spatialized" at what resolution?**

Changes: We added resolutions in the main text (see line **119**).

**L115 It would be very useful to provide the distribution of the temperature lapse-rate, even if supplement in necessary. This study from Navaro-Serrano et al. (2018) might help if you need to compare your temperature lapse-rate to similar regions.**

Changes: Done, as also recommended by reviewer 1 and 2 (see line **123ff**). Note we preferred to cite Rolland (2003) here is it specifically refers to Alpine regions.

**L118 I would suggest rewording along "An unique estimate of the precision of these data is not available as the type of sensor installed varies from one region to another. The installation and the maintenance of the sensors..."**

Changes: Done (see line **105ff**).

**L122 "remapped" quite vague. Cropped?**

Changes: Changed as recommended (see line **129**).

**L124 "each region to tailor" unclear. What is the exact meaning of "region" here? What is tailoring S3M?**

In Italy, each administrative region (first-level administrative divisions of the Italian Republic) has civil-protection offices that have access to real-time outputs of S3M Italy. Each of these administrative regions may potentially ask for region-specific parametrizations.

Changes: While the above was the intended meaning of *tailoring* in this context, we removed that passage as it was superfluous information.

**L128 "Sentinel-2"**

Changes: Done (see line **136**).

**L129 How do you manage the overlapping images? Putting on top the most recent?**

Changes: Yes, exactly. We specified this in the text (see line **138**).

**L137. "Not shown". Could be added in supplement maybe?**

Changes: We provided such statistics as a supplement for the average 2020 snow season

(see lines **148ff** and Figure S1 in the Supplement).

**L146 Please provide the number of snow depth sensor.**

Changes: Done (see line **154**).

**L151 "remapped"? unclear.**

Changes: Same as above, we used "cropped" as suggested earlier.

**L159. "For each time instant" not clear. Could be deleted.**

Changes: Done (line **174**).

**L163 What happends if snow in SCA observation but not in S3M? "preserving" is a bit unclear, maybe use "leaving without snow..."?**

Changes: Correct, we revised as suggested (see line **179**).

**L170 "The duration"?**

Changes: Rephrased (see line **185**).

**L171 "1.3 h" give it in h and min.**

Changes: Rephrased (see line **185**).

**L172 "AM" a.m.?**

Changes: Revised as suggested (see line **187**).

**L184 Given the resolution, it seems like at least the last "57" can be dropped.**

Changes: Revised as suggested (see line **199**).

**L194 Some precisions about C-SNOW product would be welcome. First, it is only available for dry snow, that is accumulation period, isn't it? Second, some part of Italy seems not covered by C-SNOW (grey area in Fig. 1 of Lievens et al., 2019). Finally, is C-SNOW completely independant from IT-SNOW? C-SNOW was calibrated on snow depth from AWS.**

Correct, C-SNOW is only available for dry snow. It is also correct that only mountain regions are covered by the product. Finally, the reviewer is right that some (but not all) snow-depth data from Italian AWS were used in developing C-SNOW, along with a variety of other snow-depth data from the Northern Hemisphere. In any case, C-SNOW remains the only, high-resolution and temporally dense remote-sensing product of snow depth that can be used to validate IT-SNOW across the Italian mountain ranges.

Changes: We added the above to the manuscript (see lines **209**).

**L211 "ASL" a.s.l.?**

Changes: Revised as suggested (see line **230**).

**L232 Is it not possible to make it "3.2 Results" and then sub-sections (3.2.n) for the different data sources?**

Changes: Revised as suggested (see e.g. line **254**).

**L235 how did you compute the RMSE? Between time-series at each pixel? Please write the MAD from Lievens et al. (2019).**

Yes, we computed RMSE between time-series at each pixel.
Changes: We added the above and the requested MAD to the manuscript (see lines **256ff**).

**L243 Please provide values for the bias. A table summing all the statistics evaluation would be really helpful.**

Changes: Revised as suggested (see lines **256ff** and Table 1).

**L248 Then there is little information brought by the comparison of IT-SNOW density with station density since the density is derived from snow depth and SWE and snow depth and SWE are also compared to IT-SNOW.**

This is correct, but at the same time we think it is important to provide error estimates for density given that this is one of the four data layers provided by IT-SNOW.
Changes: No change.

**L253 Please provide bias values.**

Changes: Revised as suggested (see Table 1).

**L284 "102 gauge stations"?**

Changes: Revised as suggested (see line **318**)..

**L294 "Again,..."**

Changes: Revised as suggested (see line **329**).

**L306 "evalution results"=> "the results"**

Changes: Revised as suggested (see line **341**).

**L307-309 Cut this long sentence int two.**

Changes: This sentence was removed as it was a repetition (see line **340**).

**L313 "peripheral"? geographicaly peripheral? The Alps are on the periphery of Italy but the station density is high...not clear.**

We meant areas that are at the periphery of the Italian territory. While station density in

these regions may remain locally high, spatially distributed information like blended gauge-radar precipitation will have an inherently larger uncertainty (because of a lack of, e.g., available radar information outside the Italian territory).

Changes: We elaborated on the above in the manuscript (see lines **344ff**).

**L317 not clear if talking about the SCA of the Sentinel-2/MODIS product or from IT-SNOW.**

Changes: We meant SCA from satellites. Clarified (see line **352**).

**L331-334 sentence is too long. To be cut.**

Changes: Clarified (see lines **361ff**).

**L334 "apriori appear"? please rephrase.**

Changes: We deleted "a-priori" as it was confusing (see line **366**).

**L336 "Thus, quantifying this uncertainty in still elusive at this stage." I dont understand where this conclusion stems from.**

This conclusion stems from previous work, such as Magnusson et al. (2015) or Avanzi et al. (2016), showing that solving the energy balance is no key driver of model performance for snow bulk variables at daily scale.

Changes: We clarified this point (see lines **361ff**).

**L336: "in"=>is**

Changes: Revised as suggested.

**L337 To move earlier in the description of the data or method.**

Changes: Revised as suggested (see lines **197ff**).

**L344 "basic science"? Please reformulate.**

Changes: Revised as suggested (see line **373**).

**L348 I like the catchy quesions. However, "what is it doing?" is not so clear and it does not appear in the conclusion. Could it be deleted? I also suggest more detailed formulation "How much is accumulated in total? Where/how is it spatially distributed?"**

Changes: Revised as suggested (see line **378** and **405**).

**L353 1st**

Changes: Revised as suggested (see line **382**).

**L353 "(this finding is in agreement with a recent reconsideration of this conventional date, see Montoya et al., 2014)" get this sentence out of the brackets**

Changes: Revised as suggested (see line **382**).

**L364 "anecdotal data" unclear if they are data from IT-SNOW or non-scientific data.**

These data come from newspapers and/or other publications, as reported in the main text.
Changes: We revised text here and removed the word "anecdotal" (see line **393**).

**L366 "150+ cm of fresh snow in 24 hours" give the date of this event.**

Changes: We were not able to determine event date from the cited literature, so we removed this information (see lines **390ff**).

**Figure 2. Provide title in the figure for each subplot, next to (a,b,c). Provide more values for the colorbar of b and c. In a, is there no station with several type of measurements? Or are they hidden because points overlapp?**

Changes: Revised as recommended (see now Figure 2).

**Make b colorbar symetric so that 0° is linked to the yellow color. In b, keep the same precision (14,8//-16,77) and meaningful values (-10,0,10. . . ).**

Changes: We tried to change the color scale to make it symmetric, but this made it difficult for one to appreciate patterns. So we kept an asymmetric color scale here. At the same time, we modify the legend so that all numbers have now same precision (see now Figure 2).

**In the legend: "by S3M Italy and thus the IT-SNOW reanalysis" => "by S3M Italy to produce IT-SNOW." Confusing otherwise.**

Changes: Revised as recommended (see now Figure 2).

**Scale is missing as well as xlabel and ylabel.**

Changes: Revised as recommended (see now Figure 2).

**If you find a way to make all subplots fits in only one line, it would make better use of the space.**

Changes: We preferred to keep them on two lines to make them as readable as possible.

**Figure 3 See comments for Figure 2 that can be applied here. For b, why is there a transparent area without data which is not of the colour of the NaN provided in the legend.**

Changes: We revised the figure and fixed the blank vs. NaN issue (see now Figure 4).

**Figure 4 This figure is much more readable. Add xlabel, ylabel to the maps and make sure color scale is symetric centered on 0. Improve the colorbar (see above).**

Changes: Revised as recommended (see now Figure 5).

**Figure 5 Suggestion for future figures: b and c would be better plotted with a heat-map or**

**at least some transparency of the points.**

    Changes: Agreed.

**Figure 8 Make the color scale continuous, it is really hard to read the map otherwise.**

    Changes: This comment was unclear to us: in our understanding the color scale is indeed continuous.

**References**

Avanzi, F., De Michele, C., Morin, S., Carmagnola, C.M., Ghezzi, A., Lejeune, Y., 2016. Model complexity and data requirements in snow hydrology: seeking a balance in practical applications. Hydrological Processes .

Magnusson, J., Wever, N., Essery, R., an A. Winstral, N.H., Jonas, T., 2015. Evaluating snow models with varying process representations for hydrological applications. Water Resources Research 51, 2707 – 2723.

Rolland, C., 2003. Spatial and Seasonal Variations of Air Temperature Lapse Rates in Alpine Regions. Journal of Climate 16, 1032–1046. doi:`10.1175/1520-0442(2003)016<1032:SASVOA>2.0.CO;2`.